# Anthropogenic Influence on the Rhine water temperatures

Alex Zavarsky[1] and Lars Duester[1]

[1]Federal Institute of Hydrology, Department G4- Radiology and Monitoring, Koblenz Germany

**Correspondence:** Alex Zavarsky (zavarsky@bafg.de)

**Abstract.** River temperature is an important parameter for water quality and an important variable for physical, chemical and biological processes. River water is also used by production facilities as cooling agent. We introduced a new way of calculating a catchment-wide air temperature using a time-lagged and a weighted average. Regressing the new air temperature vs river water temperature, the meteorological influence and the anthropogenic heat input could be studied separately. The new method was tested at four monitoring stations (Basel, Worms, Koblenz and Cologne) along the river Rhine and lowered the root-mean-square error of the regression from 2.37 $^{o}C$ (normal average) to 1.02 $^{o}C$. The analysis also showed that the long-term trend (1979-2018) of river water temperature was, next to the increasing air temperature, mostly influenced by decreasing nuclear power production. Short-term changes on time scales < 5 y were connected with changes in industrial production. We found significant positive correlations for the relationship.

*Copyright statement.* TEXT

## 1 Introduction

River water temperature ($T_w$) greatly influences the most important physical and chemical processes in rivers and is a key factor for river system health (Delpla et al., 2009). $T_w$ also confines animal habitats (Gaudard et al., 2018; Isaak et al., 2012; Durance and Ormerod, 2009), regulates the spread of invasive species (Wenger et al., 2011; Hari et al., 2006) and is therefore an important ecological parameter. River water is not solely important from an environmental perspective but is also of very significant interest for economy. Especially for energy intensive industries such as power plants, oil refineries, paper or steel mills, river water is an important resource. Its availability is a basic requirement for the facilities location (Förster and Lilliestam, 2010). As a cooling agent, given a 32 % energy efficiency, 68 % of the energy is discharged through the cooling system into the respective stream (Förster and Lilliestam, 2010). This leads to a significant heat load even on large rivers such as the Rhine (IKSR, 2006; Lange, 2009). As a consequence, anthropogenic effects such as industrial heat input, river regulation or stream-side land change can contribute significantly to the heat budget of a river and furthermore on $T_w$ (Cai et al., 2018; Gaudard et al., 2018; Råman Vinnå et al., 2018). The natural influences on $T_w$ are: [1] Meteorology, including sensible heat flux, latent heat flux, radiative heat fluxes; [2] source temperature, which describes the origin of the water, e.g. snow-fed, glacier-fed, groundwater-fed; [3] hydrology, which influences the water temperature through the amount of water and the flow velocity;

together with the change in riparian vegetation; [4] ground heat flux.

Dependent on data availability, computing power, accuracy and the questions asked, $T_w$ can be modeled in different ways. The common options are statistical models and physical based models.

A physical $T_w$ model (Sinokrot and Stefan, 1993) usually parameterizes or estimates the meteorological and ground heat fluxes and adds anthropogenic heat input. Each modeled heat flux is then applied to the water mass, initialized with the starting and

boundary conditions of source temperature and discharge. However, it is difficult to get a good estimation of these different terms over a larger catchment area. Hybrid models are in between physical based and statistical models. They use physical formulation of fluxes but determine their parameters stochastically (Piccolroaz et al., 2016). Hybrid models can reproduce river water temperatures better than simple statistical models ( e.g. linear regression) (Toffolon and Piccolroaz, 2015). Their approach includes more parameters and thus, is more complex. However, a simple hybrid model with three parameters is

comparable to a statistical model with the same number of parameters. Statistical models use air temperature ($T_a$) as a proxy for sensible, latent and radiative heat fluxes (ground heat flux can be neglected) and establish a $T_a \rightarrow T_w$ relationship through regression. $T_a$ is rather easily available from meteorological networks or reanalysis products. This is an established method and depending on the complexity, linear or exponential models are used (Stefan and Preudhomme, 1993; Mohseni et al., 1998; Koch and Grünewald, 2010). Generally exponential models delivers better results with temperature extremes. However, they

lack the distinct separation between contribution to $T_w$ from anthropogenic heat input and natural influences. Using linear models, Markovic et al. (2013) show that between 81 % - 90 % of the $T_w$ variability can be described by $T_a$. Furthermore, the authors showed that 9 % - 19 % can be attributed to hydrological factors (e.g. discharge). The study was conducted for the Danube and Elbe basins using data from 1939 - 2008. These two rivers have comparable discharges and catchment areas to the Rhine river, which could mean his results are transferable. These, although simple, linear models are able to clearly separate

the different influences on $T_w$. Another development are spatial statistical models. They correlate various landscape variables (e.g. elevation, orientation, hill shading, river slope, channel width, ...) across the catchment area and aim to statistically determine their influence on $T_w$ at a certain point. These correlations can be across any distance and do not have to satisfy flow connection or direction in the river system. As a prerequisite, a detailed knowledge about the river system and its characteristics is needed (Jackson et al., 2017a, b). An improvement to spatial statistic models was to recognize rivers as a network of

connected segments with a definite flow direction (Hoef et al., 2006; Hoef and Peterson, 2010; Isaak et al., 2010; Peterson and Hoef, 2010; Isaak et al., 2014). Correlation of the variables (e.g. $T_a$, $T_w$ discharge, ...) which influence other $T_w$, is weighed on their flow connectivity and euclidean distance or flow distance. These models can also include time-lag considerations using temporal auto correlation (Jackson et al., 2018). Artificial Neural Networks (ANN) are a subset of the statistical models and used when an incomplete understanding of most contributing processes is given (Hassoun, 1995). ANN use a sample data-set

to train artificial neurons the relationship between input (e.g. air temperature) and output ($T_w$) (Zhu et al., 2018).

We used a simple linear regression model (transferable to other streams) to investigate the temperature changes of the Rhine river over 40 years, which had been influenced by 12 nuclear power plants (NPP) along the river Rhine. These NPPs had caused, for decades, the largest part of anthropogenic heat input (Lange, 2009). The nuclear power production increased in the 1970s and 1980s and reached a peak in the mid 1990s. After the Fukushima disaster (2011), the German government de-

cided to exit nuclear power production and first NPPs were shut down. After this political decision a distinct drop on nuclear power production was visible, on top of already decreasing production rates. By July 2019 eight NPPs remained operational in the catchment area of the Rhine using (partly) river water as cooling agent. In this publication, we hypothesized that, next to environmental factors, the long-term decrease of power production, which is coupled to a decreasing use of river water as cooling agent, has a long-term (> 10 y) impact on $T_w$ of the Rhine. Short-term economic changes, observable in the change

of the gross domestic product (GDP), may influence $T_w$ on shorter time scales (< 5 y). As several industrialized hot-spots are present along the river, this impact might be spatially heterogeneous. Using the nuclear power production and GDP data, we also investigated the varying anthropogenic impact on $T_w$ along the Rhine at four monitoring stations (Basel, Worms, Koblenz and Cologne).

## 2   Methods

We investigated the change in anthropogenic heat input and its spatial and temporal heterogeneity along the Rhine combining ideas from spatial correlation models to develop a new method of calculating a representative catchment air temperature ($T_c$). $T_c$ and discharge at the measurement station $Q$ were used in a multiple linear regression $T_c \rightarrow T_w$ (Eq. 1). The resulting regression coefficients $a_1$, $a_2$ and $a_3$ describe the magnitude of the respective influences (anthropogenic heat input, meteorological and hydrological).

$$T_w = a_1 + a_2 \cdot T_c + a_3 \cdot Q \qquad\qquad(1)$$

Using an improved calculation method for $T_c$, which includes catchment-wide averaging with river-size weighing and a time-lag, the regression should deliver a better estimate for $a_1$, $a_2$ and $a_3$.

The model was run on a $T_w$ time series from 1979 to 2018 measured at four Rhine stations (Basel (CH), Worms (DE), Koblenz (DE) and Cologne (DE)). From 1979 to 2018 several changes in anthropogenic heat input to the Rhine catchment area occurred,

making it an interesting data-set to be studied. Webb et al. (2003); Markovic et al. (2013) have shown that $Q$ is inversely related to $T_w$ and an important factor in the $T_c \rightarrow T_w$ relationship. Additionally, it may function as a measure of how fast a the water mass responds to changes in $T_w$. Ground heat flux, ground water influx and heat generation due to friction were not included in this model because of the comparable small influence (Sinokrot and Stefan (1993) for the Mississippi; Caissie (2006) as review article).

Using the multiple regression (Eq. (1)), we especially investigated the change of $a_1$ over time, which we call in this study the Rhine base temperature (RBT). This temperature represents $T_w$ without the influence of meteorology ($T_a$) and discharge ($Q$). RBT was defined to be an indicator for industrial heat input and the use of Rhine water as cooling agent, in case both are mostly independent of $T_a$ and $Q$.

| station | stream km | time period | important tributary upstream | data-provider |
|---------|-----------|-------------|------------------------------|---------------|
| Basel | KM 171 | 1.1.1977-31.12.2018 | Aare | BAFU (2019) |
| Worms | KM 443 | 1.1.1971-31.12.2018 | Neckar | LfU (2019) |
| Koblenz | KM 590 | 1.1.1978-31.12.2018 | Main | BfG (2019) |
| Cologne | KM 688 | 1.1.1985-31.12.2018 | Mosel | WSA (2019) |

**Table 1.** Monitoring stations used in this study from Switzerland (Basel) to the lower Rhine region (Cologne, Germany). The location as Rhine km, the time-period, the important upstream tributary and the data-source are listed.

## 2.1 Water temperature and discharge

We used a data-set of daily averaged $T_w$ and $Q$ from 1979-2018 provided by (WSA, 2019; BfG, 2019; LfU, 2019; BAFU, 2019). The original data-sets had a 10 min sample frequency and were averaged to a daily output. Table (1) lists the respective stations along the Rhine, stream km, data availability, the important tributaries upstream and the data-provider. $T_w$ was measured by platinum resistivity sensors (Pt100). The accuracy of theses sensors is commonly $\pm 0.5$ $^o$C but the precision, which describes the ability to detect temperature changes, is 0.05 $^o$C. As we focused on the change of $T_w$ over time and did not compare the absolute temperature, the accuracy was not essential and the precision of the sensors was sufficient for this study. Measurement uncertainties (e.g. depth and location of the sensor) were not influencing the calculation, regarding the aim of this study, as long as the measured $T_w$ was a linearly dependent proxy for the average river temperature. $Q$ was provided as daily averages in m$^3$ s$^{-1}$ by the source in Tab. (1) and usually calculated from a river stage nearby.

The original data-sets were provided by state and federal operated monitoring stations which usually run backup measurement systems. They verified the data and we additionally screened the data-set for suspicious features. Missing data points up to one week were linearly interpolated. Longer or recurring data-outages were not given.

## 2.2 Air temperature

$T_a$ is retrieved from the European Centre for Medium-Range Weatherforcast (ECMWF) Reanalysis Model ERA5. It provides an hourly time resolution of the 2 m $T_a$ on a $\frac{1}{4}^o$ by $\frac{1}{4}^o$ grid. The data-set is available from 1979-2018. We took the hourly $T_a$ output and calculated a daily mean for each grid point between 1979 and 2018 to fit the time resolution of $T_w$.

## 2.3 Nuclear Power Plants

The annual electrical power production (EPP) by NPPs is available from the International Atomic Energy Agency (IAEA) Power Reactor Information System (IAEA, 2019). 12 NPPs (1986-1988) were online in the Rhine catchment area and eight remained operational by July 2019. All shutdowns were undertaken in Germany. In this study separate reactor blocks of the same plant NPP were combined.

The heat input (HI) by NPPs to the Rhine was calculated for each monitoring station using the conversion factor $c$ and the yearly EPP, Eq. 2. NPPs with an exclusive river water cooling system have a conversion factor of three, which is based on

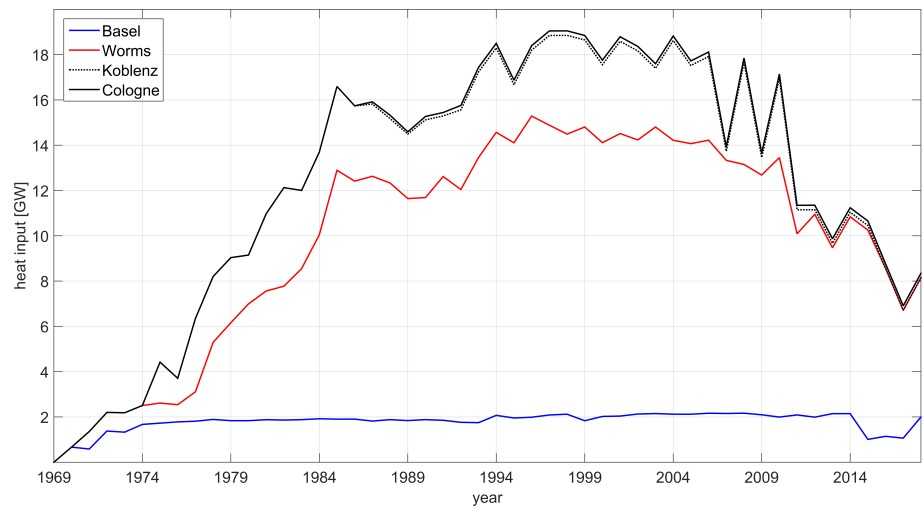

**Figure 1.** Heat input by upstream NPP from 1969 to 2018 at each monitoring station.

| NPP | country | river | conversion factor | const. HI |
|---|---|---|---|---|
| Beznau I+II | CH | Aaare | 3 | N/A |
| Biblis I+II | DE | Rhine | 2 | N/A |
| Cattenom I-IV | DE | Mosel | N/A | 200 MW |
| Fessenheim I+II | FR | Rhine | 3 | N/A |
| Goesgen | CH | Aare | N/A | 50 MW |
| Grafenrheinfeld | DE | Main | N/A | 200 MW |
| Leibstatt | CH | Rhine | N/A | 50 MW |
| Muehleberg | CH | Aare | 3 | N/A |
| Neckarwestheim I+II | DE | Neckar | 1 | N/A |
| Obrigheim | DE | Neckar | 3 | N/A |
| Philippsburg I+II | DE | Rhine | 1 | N/A |

**Table 2.** NPPs included in this study. The conversion factor describes the conversion from EPP to HI. If cooling towers are installed a constant heat input was used for the calculation based on Lange (2009).

the power efficiency of electricity generation (Lange, 2009). Other factors are estimated depending on the cooling system and personal communication. If no conversion factor was available a constant HI was assumed (Lange, 2009).

115
$$HI[GW] = \frac{c \cdot EPP[GWh]}{365 \cdot 24[h]} \qquad (2)$$

The NPPs, their conversion factor and if applicable the constant HI are shown in Tab. 2. The time series of upstream HI by NPPs for each monitoring station is shown in Fig. 1.

**Calculated temperature change**

We calculated the expected change $\Delta T_w$ based on a change in HI ($\Delta$ HI) by NPPs using the average discharge $\bar{Q}$, the heat
capacity of water $c_p$ and the water density $\rho$, Eq. (3).

$$\Delta T_w = \frac{\Delta HI}{c_p \cdot \bar{Q} \cdot \rho} \tag{3}$$

This approach follows the idea that the contribution of NPPs significantly alters the $T_w$ and only influences the RBT fraction.

## 2.4 Gross domestic product

The GDP for the adjacent German federated states is obtained from VGdL (2019a, b). Due to changes in the calculation method
of the GDP before and after the German reunification (1990), two separate data-sets were used. For this study only the GDP-
change of the secondary sector (construction and production) was taken into account.

The RBT, if compared to the GDP, was filtered using a $10^{th}$ order butterworth bandpass filter. The sampling rate of the GDP
was 1 y$^{-1}$. We used 1.1 y$^{-1}$ as higher and 0.05 y$^{-1}$ as lower cutoff frequencies for RBT. This means that signals with a
periodicity larger than 20 y and lower than 0.9 y were excluded from calculations and display. The reasoning was to make the
RBT data comparable to the yearly data of the GDP-change. The low frequency cutoff was canceling long-term trends as the
GDP-change was only related to the previous year. The high frequency cutoff was used to dampen fast alternating RBT signals
in comparison to the slow sampled GDP data.

## 2.5 Rescaled adjusted partial sums

Rescaled adjusted partial sums (RAPS) were used to visualize trends in time series which may not be clearly visible in the
unprocessed data-set. Equation (4) shows the calculation of the RAPS index $X$ using a time series Y.

$$X_k = \sum_{i=1}^{i=k} \frac{Y_i - \overline{Y}}{\sigma_Y} \tag{4}$$

$\overline{Y}$ is the average over the total time series, $\sigma$ is the standard deviation of the whole time series, $Y_i$ is the i$^{th}$ data-point in $Y$.
A change in the slope of the RAPS index only indicates a change in the slope of the original time-series. A negative RAPS
slope does not indicate a negative slope in the original time series. Garbrecht and Fernandez (1994) and Basarin et al. (2016)
used this method to investigate trends in hydrological time series.

## 2.6 Catchment area

The catchment area was calculated using the Hydrosheds database (Lehner et al., 2008). The $\frac{1}{125}^o$ by $\frac{1}{125}^o$ gridded data-set
provides information, at each grid point, to which cell the water of a grid cell is drained. By selecting a starting location, e.g.
Koblenz at $50.350^o$ N and $7.602^o$ E it was possible to iteratively identify all grid points draining into this location. These grid
points represent the catchment area of this location (in the example from Fig. (2) Koblenz). By counting the iteration steps,

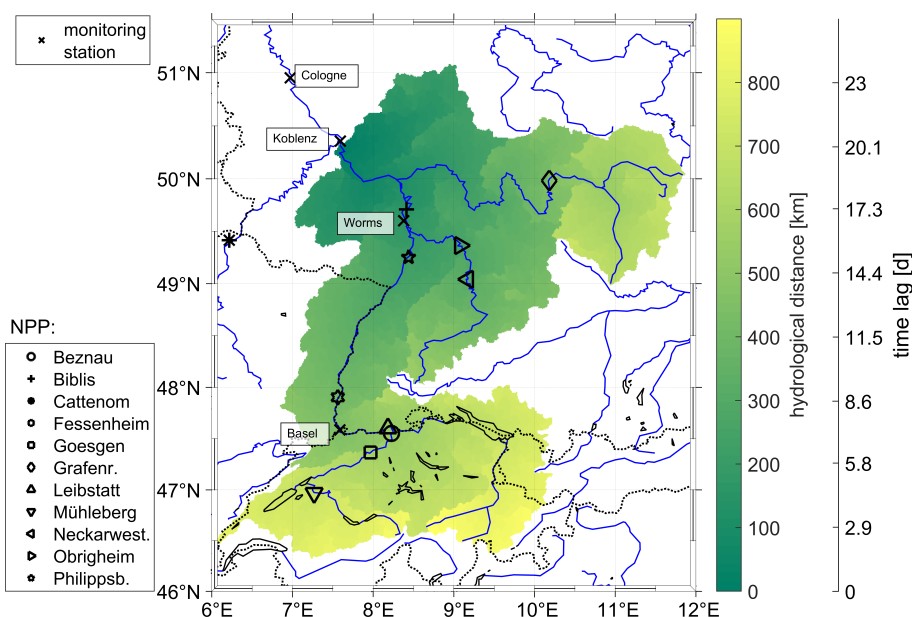

**Figure 2.** Catchment area of the Koblenz monitoring station as an example. The colors show the hydrological distance between the monitoring station and each grid point of the catchment area. The second y-axis shows the time it takes in days, in our set-up, to flow from a certain grid point to the monitoring station based on the hydrological distance. The flow speed is $0.4 \text{ m s}^{-1}$ and was defined to be constant in space and time. All monitoring stations are marked by X. The other markers show the location of the NPPs.

the distance a water drop travels to reach the monitoring station Koblenz was determined. This was done for each of the four stations. Additionally, the accumulation number $ACC$ was obtained from the data-set. It defines how many cells in total were draining into a particular cell and it is a measure for the size of a river. Finally, a grid, which defines the catchment area, the $ACC$ and the hydrological distance was established spanning the whole catchment area. Figure (2) shows the catchment area,

the hydrological distance and the calculated flow time to the Koblenz monitoring station.

 The $ACC$ displays is the number of grid points which were hydrologically connected to this specific grid point. Figure 3 (top panel) shows the distribution of the $ACC$. Large rivers, which have a large $ACC$ number, such as the Rhine, Main, Neckar are easily visible due to their green to yellow color.

## 2.7 Multiple regression

A multiple linear regression was used to separate the anthropogenic heat input $a_1$, meteorological $a_2$ and hydrological $a_3$ contributions to the river water temperature. $T_w$ was regressed with $T_c$ and river discharge $Q$. Their regression coefficients $a_2$ ($T_c$ slope) and $a_3$ ($Q$ slope) represent the magnitude of the respective influences. The offset $a_1$ (RBT) combines all other influences which were not related to a change in $\text{T}_c$ or $Q$. We hypothesized that the RBT is directly linked to heat input by power plants, in this study NPPs, and other industrial facilities.

Instead of taking $T_a$ directly at the monitoring station, we improved Eq. (1) by a time-dependent and weighed average of $T_a$ (x,y,t) over the whole catchment, Eq. (5). (x,y) were spatial coordinates in the catchment area and the subscript $_0$ marks the location of the monitoring station.

$$T_w\left(x_0, y_0, t_0\right) = a_1 + a_2 \cdot T_c\left(x_0, y_0, t_0 + \Delta t\left(x, y\right)\right) + a_3 \cdot Q\left(x_0, y_0, t_0\right) \tag{5}$$

The new representative catchment temperature was called $T_c\left(x_0, y_0, t_0\right)$. It was a weighed average of the whole catchment area

(x,y) which is defined by the measurement point $(x_0, y_0)$. The difference between the measurement time $t_0$ and the reading of $T_a$ is called time-lag $\Delta t(x, y)$ and depends on the hydrological distance between the measurement point and the reading. $\Delta t(x, y)$ is negative and points to a moment in time before the measurement.

### 2.7.1   Time-lag

A change in $T_w$ is slower than a change in $T_a$. The time-lag $\Delta t$ describes this lagging and is commonly used in water temper-

ature models.

A reason for the occurrence of $\Delta t$ is that the water mass' mixing capability, heat capacity and surface area cause a strong thermal inertia. Changing $T_w$ through new meteorological conditions and heat fluxes take time. Therefore, linear as well as exponential models include either a fixed $\Delta t$ for $T_a$ (Eq. 6) or an average of $T_a$ including a time span before (Eq. 7) (Stefan and Preudhomme, 1993; Webb and Nobilis, 1995, 1997; Haag and Luce, 2008; Benyaha et al., 2008).


$$T_c\left(x_0, y_0, t_0\right) \quad = \quad T_a\left(x_0, y_0, t_0 + \Delta t\right) \tag{6}$$

$$T_c\left(x_0, y_0, t_0\right) \quad = \quad \sum_{t=t_0}^{t=t_0+\Delta t} T_a\left(x_0, y_0, t\right) \tag{7}$$

A second reason for a mismatch is advection. Rivers, in this case the Rhine, exhibit current velocities which enable its water to cover significant distances on time scales larger than days. Therefore, it is necessary to take the change of $T_a$, in space and

time, during advection into account. This is especially important for daily averaged $T_w$ (Erickson and Stefan, 2000). Pohle et al. (2019) average eight days of hydroclimatic variables over the whole catchment area, Eq. (8). However, this approach does not include the characteristics of flow path and flow speed.

$$T_c\left(x_0, y_0, t_0\right) = \sum_{x=0, y=0, t=t_0}^{x=n, y=m, t=t_0-8} T_a\left(x, y, t\right) \tag{8}$$

We combined the general idea of a time-lag and averaging $T_a$ over the whole catchment area (Eq. 6, 7 and 8), but in this study

each grid point was linked to a specific time-lag $\Delta t\left(x, y\right)$, which is dependent on a fixed flow speed $v$ and the hydrological distance $s(x, y)$ to the measurement point, Fig. (2). The distance was obtained from the discharge map (Sec. 2.6) and calculated with $v$ as described by Eq. (9). The new $\Delta t\left(x, y\right)$ represents the mismatch by advection but not specifically the mismatch through thermal inertia. The thermal inertia would be independent of $s(x, y)$ and a constant added to $\Delta t$. However, we are of

| $\Delta t$ [d] | weighing factor | distance from measurement point [km] |
|---|---|---|
| 0 | 1 | 0 |
| -1.01 | 0.96 | 35.1 |
| -2.00 | 0.92 | 69.6 |
| -5.02 | 0.81 | 174.6 |
| ... | | |
| -13.01 | 0.50 | 452.5 |
| ... | | |
| -26 | 0 | 904 |

**Table 3.** Weighing factors for the distance and the resulting $\Delta t$ for the monitoring station Koblenz. $\Delta t$ is calculated from distance and flow speed, Eq. (9). The weighing coefficient is linearly correlated to the $\Delta t$.

the opinion, that a sufficient part of the thermal inertia time-lag was included in our representation of $\Delta t(x,y)$.

$$\Delta t(x,y) = -\frac{s(x,y)}{v} \tag{9}$$

### 2.7.2 Weighing coefficients

Tobler (1970) proposed that close spatial and temporal conditions tend to be higher correlated than those further away. This led to the introduction of the weighing factor $w$. A linear decreasing weighing factor from 1 to 0 was used. 1 marks the grid point closest (smallest $\Delta t$) to the monitoring station and 0 the point farthest away (largest $\Delta t$). As the size of the catchment areas were different for the four monitoring station, four weighing coefficient tables were calculated. As an example, Table (3) shows the weighing coefficient for Koblenz.

A catchment-wide hydrological flow model, estimating the flow speed at every grid point for every hydrological scenario, was not used. It had not been available yet for every grid point of the catchments and the the focus of this study was to create a simple set-up, also transferable to other river catchments. Therefore, it was decided to work using a constant flow speed of 0.4 m s$^{-1}$ . This flow speed was determined by calculating the RMSE (whole data-set) of the $ACC+\Delta t$ model with a step wise reduction of the flow speed from 1.4 m s$^{-1}$ to 0.3 m s$^{-1}$. The lowest RMSE was obtained at Koblenz at 0.4 m s$^{-1}$. The RMSE and NSC coefficients at all flow speeds and all stations are shown in the supplement. For Basel and for Worms slower flow speeds would lower the RMSE further. We did not include this as it would create unreasonable low flow speeds. The reason for the small flow speeds, with lowest RMSE, might be the thermal inertia. As thermal inertia is not explicitly represented in $\Delta t$ (Eq. 9) a smaller flow speed could compensate for that, especially in smaller catchment areas.

The weighing coefficient $w$ is combined with $ACC$. $ACC$ is used as a second coefficient which over-weighs grid points with large accumulation and therefore large water masses. This ensures a balance between the large number of low $ACC$ grid points, which carry less water, and the influence of $T_a$ on large water masses. Figure (3) (bottom panel) shows $ACC \cdot w$ over the whole catchment area of Koblenz.

The grid points were binned according to their $ACC$ value. A high bin represents large rivers, a low bin their tributaries. The reason was to investigate the importance of different $ACC$ bins to the total $T_c$ calculation. The $ACC$ bin with the largest contribution in Fig. (4) was normalized to one making it a relative contribution. The red bars (Fig. (4)) show the relative contribution (y-axis) of each $ACC$ bin by the number of grid points in this bin only, no $ACC \cdot w$ weighing was applied. The results showed that the large number at low $ACC$ bins (small water mass) have a larger influence compared to the rather low

numbers at high $ACC$ bins (e.g. large water masses, rivers, lakes). The difference in relative contribution is four powers of magnitude. The white bars show the relative contribution using the number of grid points in the bin and the $ACC \cdot w$ weighing. This distribution delivered rather equal importance to all grid points as it puts more weight on grid points covering lakes and rivers. The average difference in relative contribution is about one power of magnitude.

### 220   2.7.3   $T_c$

Combining $\Delta t$, $ACC \cdot w$ weighing and the gridded temperature reanalysis data of Sec. (2.2), we proposed a new 3D $(x, y, t)$ averaging of $T_a$ shown in, Eq. (10).

$$T_c(x_0, y_0, t_0) = \frac{1}{n \cdot m} \frac{1}{\sum w(\Delta t(x,y)) \cdot ACC(x,y)} \sum_{x=1, y=1}^{x=n, y=m} w(\Delta t(x,y)) \cdot ACC(x,y) \cdot T_a(x, y, t_0 + \Delta t(x,y)) \qquad (10)$$

$T_c(x_0, y_0, t_0)$ was calculated by weighed $(ACC \cdot w)$ averaging $T_a(x, y, t + \Delta t(x,y))$ over all grid points of the catchment area

(x=1,...n y=1,...m) which was set by the measurement point $(x_0, y_0)$. The time-lag $\Delta t$ was an estimate for the time it takes for a water droplet from a specific grid point (x,y) in the catchment area to the measurement location.

Based on Eq. (10), the daily $T_c$ was calculated for each monitoring station. This temperature represents the meteorological influence all water droplets have experienced on their way to the monitoring station and is subsequently used in the multiple linear regression.

### 230   2.7.4   $T_c$ calculation methods

Additionally, we used these four calculations methods [1] $w + \Delta t$; [2] avg+$\Delta t$; [3] avg; [4] point, to compare their results of the linear regression to the calculation proposed in Eq. (10).

[1] $w$ weight (Eq. 11) and time-lag only.

$$T_c(x_0, y_0, t_0) = \frac{1}{n \cdot m} \frac{1}{\sum w(\Delta t(x,y))} \sum_{x=1, y=1}^{x=n, y=m} w(\Delta t(x,y)) \cdot T_a(x, y, t_0 + \Delta t(x,y)) \qquad (11)$$

[2] No weight, only time-lag, Eq. (12).

$$T_c(x_0, y_0, t_0) = \frac{1}{n \cdot m} \sum_{x=1, y=1}^{x=n, y=m} T_a(x, y, t_0 + \Delta t(x,y)) \qquad (12)$$

[3]A mean $T_a(x_0, y_0, t_0)$ over the whole catchment area at the time $t_0$ of the measurement, Eq. (13). $\Delta t$ was not used.

$$w(x,y) = 1 \qquad\qquad T_c(x_0, y_0, t_0) = \frac{1}{n \cdot m} \sum_{x=1, y=1}^{x=n, y=m} T_a(x, y, t_0) \tag{13}$$

[4]$T_a(x_0, y_0, t_0)$ at the location $x_0, y_0$ and time $t_0$ of the measurement, Eq. (14).

$$T_c(x_0, y_0, t_0) = T_a(x_0, y_0, t_0) \tag{14}$$

## 3 Results

### 3.1 Water temperature time series

To investigate the long-term change over time, we fitted a time dependent linear function to the time series of $T_w$ and $T_a$ (catchment average) of all four monitoring stations (Basel, Worms, Koblenz and Cologne). The same was also done, when all four monitoring stations had an overlapping data-set (1985-2018), Tab. (1). The left column of Fig. (5) presents the yearly averaged $T_w$ and the linear fits for the two time periods. The average $T_a$ of the catchment area is also shown. In the right column of Fig. (5) the RAPS index of $T_a$ as well as $T_w$ is shown. The fit coefficients and the rate of warming per year are displayed in Tab. (4).The calculated $T_a$ increased in the catchment area of all monitoring stations and the respective slopes are shown in column four and five of Tab. (4).

Figure (5) and Table (4) show that the change of $T_w$ was found to be heterogeneous along the Rhine. The slope at Basel is approx. six times higher (0.049 $^o$C y$^{-1}$) than the one in Cologne (0.0084 $^o$C y$^{-1}$), comparing only the overlapping data-set. However, during the same period $T_a$ (0.05 $^o$C y$^{-1}$ Basel, 0.05 $^o$C y$^{-1}$ Cologne) display similar behavior at these two stations, which is an indication of similar meteorological influence. The $T_w$ warming rate from 1985-2018 for Worms and Koblenz are in between those from Cologne and Basel. These two stations show similar $T_a$ warming rates compared to Basel and Cologne. Generally, the $T_a$ warming rates are less than 5 % different from each other. Arora et al. (2016) showed a mean $T_w$ warming rate of north and north-east Germany rivers of 0.03 $^o$C y$^{-1}$ (1985-2000) and 0.09 $^o$C y$^{-1}$ (2000-2010). Regarding our time-period (1985-2010) these values are plausible. Basarin et al. (2016) found a maximum increase of $T_w$ at the Danube at Bogojevo (1950-2012) of 0.05 $^o$C y$^{-1}$ which is matching the maximum increase at Basel. $T_a$ increased by 0.02 $^o$C y$^{-1}$ between 1985-2010 in the study by Arora et al. (2016). We found a steeper slope at all stations. The reason could be the hiatus of global warming (Hartmann et al., 2014), which is a flattening of the $T_a$ increase between 1998-2012. This period is fully included in the Arora et al. (2016) and our data-set but we investigated further until 2018, when the warming of $T_a$ has already increased again (Hu and Fedorov, 2017). Michel et al. (2020) investigated $T_w$ at 52 river gauges in Switzerland representing most of the Rhine catchment area at Basel. The authors reported an average $T_w$ increase at the 52 stations of 0.037 $^o$C y$^{-1}$ (1998-2018) and 0.033 $^o$C y$^{-1}$ (1979-2018). $T_a$ increased 0.039 $^o$C y$^{-1}$ (1998-2018) and 0.046 $^o$C y$^{-1}$ (1979-2018). Comparing this to our results at Basel, the T$_a$ warming rates are similar. The difference might originate from the use of meteorological stations nearby river gauges only (Michel et al., 2020) instead of a reanalysis product. The difference of $T_w$ warming (approx. 0.021 $^o$C y$^{-1}$) could be interpreted that a lot of warming might occur in the broader vicinity before the Basel monitoring station.

| station | slope $T_w$ whole data-set [$^oC$ y$^{-1}$] | corr. $T_w \leftrightarrow T_a$ whole data-set | slope $T_w$ 1985-2018 [$^oC$ y$^{-1}$] | corr. $T_w \leftrightarrow T_a$ 1985-2018 | slope $T_a$ whole data-set [$^oC$ y$^{-1}$] | slope $T_a$ 1985-2018 [$^oC$ y$^{-1}$] |
|---|---|---|---|---|---|---|
| Basel | $0.054, R^2 = 0.66$ | 0.867 | $0.049, R^2 = 0.38$ | 0.874 | $0.050, R^2 = 0.48$ | $0.050, R^2 = 0.32$ |
| Worms | $0.055, R^2 = 0.52$ | 0.690 | $0.035, R^2 = 0.38$ | 0.729 | $0.050, R^2 = 0.20$ | $0.048, R^2 = 0.36$ |
| Koblenz | $0.033, R^2 = 0.31$ | 0.778 | $0.024, R^2 = 0.38$ | 0.762 | $0.052, R^2 = 0.11$ | $0.048, R^2 = 0.36$ |
| Cologne | $0.008, R^2 = 0.001$ | 0.499 | $0.008, R^2 = 0.31$ | 0.499 | $0.050, R^2 = 0.001$ | $0.050, R^2 = 0.31$ |

**Table 4.** Slope of linear fits and Pearson's correlation coefficients to the daily temperature data at the four monitoring stations. The data-set used is described in the column header. Next to the slope values are the $R^2$ values, which are statistically significant if $R^2 > 0.19$

The $R^2$ values make differences between the four monitoring stations visible. Basel exhibits the largest $R^2$ values and these are consistently high for $T_a$ and $T_w$. This is in contrast to the station Cologne, where $R^2$ of $T_w$ was low and statistically not significant. The slope of $T_a$ at Cologne is lower than at the other stations but still statistically significant. The Pearson's correlation coefficients between $T_a$ and $T_w$ were lowest at Cologne and largest in Basel. For $T_a$ the RAPS index of all monitoring stations showed four concurrent sections (start-1987; 1987-2000; 2000-2014; 2014-end). Their borders are marked by the blue triangles in Fig. (5). The section between 2000-2014 could be a consequence of the hiatus of global-warming between 1998-2012 (Hartmann et al., 2014). Each section represent slope changes of the RAPS index and indicate trend changes in the original time-series. The $T_w$ RAPS index for Basel displayed the same pattern of sections as the $T_a$ index. All other stations showed other RAPS $T_w$ to RAPS $T_a$ patterns. This means that the $T_a$ and $T_w$ trends of the original time-series were different at these stations. $T_a$ can not fully describe the trends in $T_w$.

We hypothesized that different meteorological conditions were not the reason for this difference. Meteorological differences should have also been visible in the $T_a$ warming rate of the four stations, which was not the case. The RAPS analysis for $T_a$ and $T_w$ only coincided within the Basel data-set.

## 3.2 Regression

We fitted the multiple regression model (Eq. 5), using $T_c$ and $Q$ to $T_w$ of each monitoring station for the available data-set. Afterwards, we recalculated $T_{w,modelled}$ using the regression coefficients $a_1$, $a_2$ and $a_3$. From the comparison between the $T_{w,modelled}$ and measured $T_w$, the root mean square error (RMSE) and the Nash-Sutcliffe coefficient (NSC) for each monitoring station was derived, Tab. (5). To support the introduction of weighing coefficients $ACC \cdot w$ and $\Delta t$, we compared five different calculations of $T_c$ from Sec. (2).

Table (5) shows the RMSE and NSC values for all correlations. The lowest (RMSE) and highest (NSC) values were displayed bold in Tab. (5). The lowest RSME was found to be 1.02 $^oC$ for $ACC \cdot w + \Delta t$ (row one) at the Koblenz station. At this location also the largest NSC of 0.97 appeared. The flow speed was optimized for lowest RMSE at the Koblenz station, Sec. (2.7.2). It was evident that the three methods including a $\Delta t$ have a lower RMSE (below 2.01 $^oC$, lowest 1.02 $^oC$) than the two methods without a $\Delta t$ (above 2.37 $^oC$, largest 2.97 $^oC$). The same trend held true for NSC where the $\Delta t$ methods were above 0.90 and the other two were below 0.86. We think that the use of a catchment-wide $\Delta t$ improved the quality of the multiple

| method | RMSE | | | | NSC | | | |
|---|---|---|---|---|---|---|---|---|
| | Basel | Worms | Koblenz | Cologne | Basel | Worms | Koblenz | Cologne |
| $ACC \cdot w + \Delta t$ | 1.65 | **1.24** | **1.02** | **1.31** | **0.93** | **0.96** | **0.97** | **0.95** |
| (1) w+$\Delta t$ | **1.56** | 1.33 | 1.43 | 1.87 | 0.92 | 0.95 | 0.95 | 0.92 |
| (2) avg+$\Delta t$ | 1.61 | 1.45 | 1.70 | 2.08 | **0.93** | 0.94 | 0.93 | 0.90 |
| (3) avg | 2.48 | 2.43 | 2.37 | 2.97 | 0.82 | 0.84 | 0.86 | 0.79 |
| (4) point | 2.67 | 2.55 | 2.63 | 2.85 | 0.78 | 0.82 | 0.82 | 0.80 |

**Table 5.** RSME [$^oC$] and NSC for all $T_c$ calculation methods. Different $T_c$ calculation methods and the regressions are applied over the total data-set. The RMSE and the NSC are calculated between $T_w$ and $T_{w,modelled}$. The first column contains the calculation method. The best results for each monitoring station and each calculation method are bold.

regression analysis and delivered a significant improvement to the $T_a \rightarrow T_w$ based modeling. Interestingly, combining $ACC$ and the weighing factor $w$ provided the best estimation for all stations, except for Basel. The content of Fig. (4) could explain this result. Without $ACC$ weighing small water masses (small $ACC$) may be over-represented in the contribution to $T_c$. Large $ACC$ grid points represent large water masses (rivers and lakes) and their influence on $T_a$ may be otherwise underestimated. At Basel the fraction of low $ACC$ grid points was relatively small compared to the other stations, as Basel is closest to the water sources and has the smallest catchment area. Therefore, the $ACC$ weighing might have provided weaker results. As $ACC \cdot w + \Delta t$ provided the smallest RMSE, this calculation method was used for all further calculations of $T_c$. In the supplement we provide a calculation of the regression coefficients for the year 2001 only. These coefficients were then taken as a basis to calculate $T_w$ for each year from 2000 to 2018. The RMSE and NSC data was consistent in magnitude with the long-term regressions of this section. The RMSE at Koblenz ranged from 0.75 $^oC$ to 1.22 $^oC$. A lower RMSE was caused by the shorter regression period. This supports the stability and validity of the regression model.

### 3.3 Rhine base temperature

The RBT was taken to explain differences in the $T_w$ warming rates of Tab. (4). We regressed a two-year segment of the $T_w$ time series and set a step size of one month in order to create a RBT time series over the full data-set. The regression of a two-year segment should also compensate extreme events occurring during one year. These could be extreme low discharge or extreme water temperatures, to which industrial and power production had to react. As the absolute RBT cannot be meaningfully interpreted, only the changes of RBT over time are shown in Fig. (6). We subtracted the last data point of each time series from the rest of the data and showed the change of RBT, a four-year running mean and $\Delta$ RBT (Eq. 3) vs time. The HI by NPPs is shown as a dotted blue line with the y-axis on the right (Fig. (6)).

#### 3.3.1 Long-term trend

In this study, long-term trends were visible on time scales of decades. The HI by NPPs, the four-year running mean RBT and $\Delta$RBT followed a similar trend in this analyis, Fig. (6). After the maximum heat discharge from NPPs between 1996-1998,

| name | period | ΔRBT from data-set | $\Delta T_w$ from Eq. (3) | ΔHI [GW] |
|---|---|---|---|---|
| Basel | 2008-2017 | -0.26 | 0.04 | 0.17 |
| Worms | 1996-2017 | 1.29 | 1.19 | 7.14 |
| Koblenz | 1999-2017 | 1.59 | 1.45 | 10.5 |
| Cologne | 1998-2017 | 1.21 | 1.55 | 10.7 |

**Table 6.** Change of RBT (column three) in the time period given in column two. The start of the period indicates the maximum HI of NPPs at the respective monitoring station. The calculated $\Delta T_w$ (column four) and the change in HI by nuclear power plants (column five) are also provided. The calculations were done using Eq. (3)

the HI as well as the RBT of Worms, Koblenz and Cologne declined. The RBT started its decline 1-2 years before 1995, which might have been triggered by the recession in 1993 and a sharp drop in the German trade-balance. At Basel the RBT as well as the HI remained comparably constant. Additionally, we calculated $\Delta T_w$ based on the change in HI, using Eq. (3), at every station and compared it to the ΔRBT from the regression model, Tab. (6). The period for each monitoring station starts at the maximum HI by NPPs for the respective station and ends in the year 2017.

At Basel, both simulated and calculated RBT changes were negligible due to the lack of change in HI. At all other stations, the change in HI was reflected in the change of RBT. The maximum difference between simulation and calculation was found to be 0.34 $^oC$. Before 1995 Worms, Koblenz and Cologne showed an approx. 1 $^oC$ offset between ΔRBT and $\Delta T_w$ (Fig. (6)). This was occuring during a time when the NPPs HI remained relatively stable but the GDP increased by 30 % between 1985-1995 (Worldbank, 2020). The change in nuclear power production over a time period of 30 years or more can explain changes and the heterogenous warming rates of $T_w$ along the river Rhine. NPPs may also impact $T_w$ at much shorter time scales but do not change their power output accordingly.

### 3.3.2 Short-term trend

Short-term changes (< 5 y) in RBT (Fig. 6) are not likely to be influenced by the overall HI from NPPs, as these adopt production at longer time scales. More important are local industrial conditions, which could also include fossil fuel power plants. However, not all influences to the coefficient $a_1$ and subsequently to RBT originate from industrial production. Various potential influences are unknown and not within the scope of this publication.

For Basel, it was not possible to satisfyingly explain the short-term variations. The Rhine and its tributaries upstream are flowing through sub-alpine lakes and, in relation to the downstream part, are not strongly industrialized. Lakes have a complicated heat budget (Råman Vinnå et al., 2018), which was not focused on in this analysis.

For all other stations, we hypothesized that local production facilities and their HI into the Rhine are responsible for the short-term changes by comparing the RBT time series to economic data. Figure (7) shows the comparison of RBT (black line, one year running mean) vs the changes in the GDP (blue line). A discontinuity in the GDP at 1991 is visible, due to the German reunification, when the calculation method of the GDP changed. Therefore the data was plotted as separate lines. For Worms

(Fig. 7, bottom panel) we added the change of turnover of the BASF company (red dashed line (AG, 1989)). BASF is a major chemical company and one of its largest production facility, with an estimated HI of 500 MW to 1 GW, is located 12 km upstream (km 431) from the Worms station. It was investigated if the production and HI changes of this factory were also visible. In 1985, although the change in GDP did not indicate a large RBT change, a RBT decrease was visible. This was indicated by a turnover decrease in 1985 and 1986. After the German reunification 1990, a negative GDP change (recession)

was evident. This was followed by a BASF turnover decline as well as a decrease in RBT. After that, the RBT followed the up and down movements of the GDP and so does the BASF turnover (only shown until 2000). Especially the economic events such as the burst of the dot-com bubble (early 2000s) and the mortgage crisis (2008) were visible in the RBT and in the GDP, when a decrease of both parameters followed. The two events are marked by triangles in Fig. (7).

Before 1990, the RBT at Koblenz did not follow the GDP trend and showed a rather anti-cyclic behavior, which can not be

explained yet. After 1991, the RBT followed the general trend of the GDP but did not seem to be strongly influenced by the short recession after the German reunification. Again, economic events such as the burst of the dot-com bubble (early 2000s) and the mortgage crisis (2008) displayed an influence on the RBT.

The RBT at Cologne did not seem to be strongly influenced by the recession connected to the German reunification, but after 1999 the RBT follows the up and down trends of the GDP.

For all monitoring stations, a red dashed line was added between 1995 and 1999. This dashed line indicates the production rate of German oil refineries (MWV, 2003). From 1995 to 1999 German refineries ran at full capacity (100 %). Usually the capacity levels did not exceed 90 %. The increase in production was clearly visible in the RBT ar Cologne, where a large oil refinery is located 19 km upstream at km 671 (Rheinland refinery). The RBT at Worms and Koblenz could be influenced by the output of a refinery next to Karlsruhe at km 367 (Mineraloelraffinerie Oberrhein).

### 3.3.3 Correlation

We correlated the GDP-change and the filtered RBT signal. It was noticeable that a 480 days shift to the past was needed to get matching trends. This means that a change in RBT or anthropogenic HI appeared about 480 days earlier than in the GDP calculation. The shift could be caused by two reasons: [1] Using the GDP difference of two consecutive years, has a significance at a unspecific point of time within these two years. [2] The GDP is lagging behind the real economic situation, in this

case the industrial production. Yamarone (2012) claimed that GDP was a coincident economic indicator similar to industrial production. However, the author used quarterly GDP calculations and in this study annual data was used. The quaterly data-set may react faster to changes. A second thought was that (Yamarone, 2012) compared industrial production calculations, which is an economic index, to GDP (another economic index). In this study real-time data from industrial HI into the river was processed. This shift has not been done for Fig. (6) because a shift of 1.5 y on a 40-year time scale is negligible.

Table (7) shows the Spearman's rank correlation coefficients of Worms, Koblenz and Cologne for $ACC \cdot w + \Delta t$ calculation method, which resulted in the lowest RMSE in Koblenz. All correlations were found to be positive and statistically significant ($p<0.05$). The correlation in Koblenz was highest. Fig. 8 shows the filtered RBT signal vs the GDP-change at the three monitoring stations. The RBT time-series was detrended and filtered. Most of the time, the variations in the RBT (filterend

| name | $ACC \cdot w + \Delta t$ | significance |
|---|---|---|
| Worms | 0.48 | p<0.05 |
| Koblenz | 0.53 | p<0.05 |
| Cologne | 0.44 | p<0.05 |

**Table 7.** Spearman's rank correlations between RBT and GDP-Change for $ACC \cdot w + \Delta t$. The last column shows the significance.

and shifted) were coincident with the GDP-change. The RBT peak between 1995-1998 was not very well represented by the
GDP-change, which has already been discussed earlier in context of Fig. 7.

## 4   Conclusions

We introduced a new catchment-wide air temperature $T_c$, which decreased the RMSE (Tab. 5) in a $T_c \rightarrow T_w$ regression. $T_c$ is a weighed ($ACC \cdot w$) average of all $T_a$ across the catchment area including the use of $\Delta t$ for each grid point according to the hydrological distance and flow speed. In the approach, this time-lag was used as an indicator for the point in time when a
water droplet was at a certain grid cell in the catchment area. As a result, one can get a better estimate which $T_a$ a water droplet experienced on its way to a certain point (in this study a monitoring station) and it delivered better linear $T_c \rightarrow T_w$ estimates. This improvement in the $T_c \rightarrow T_w$ relationship may support the analysis of processes in the heat budget of rivers. Usually $T_a$ data is readily available and can easily be combined with $Q$ data for multiple linear regression analysis. Still a sufficient long (decade) time-series of $T_w$ was required. Nevertheless a linear relationship was found to be simpler than a full physical model
which requires all meteorological fluxes as input quantities.

In the prove of concept, we focused on the Rhine catchment area but in principle the model can be applied to any river system around the globe, if the respective long-term data are available. However, catchment-area data and reanalysis $T_a$ data are often globally available. Morrill et al. (2005) showed a linear $T_a \rightarrow T_w$ relationship for 43 rivers with various catchment areas in the subtropics. This potentially indicates that the proposed model and procedure can be applied elsewhere. However, this still has
to be verified. Future calculations may be coupled with catchment-wide hydrological models to improve the accuracy of the time-lag. The time-lag used in this study was based on try and error in search for the lowest RMSE. A detailed catchment wide hydrological flow model would be especially beneficial to set an upper limit for the time-lag and constrain its validity. It would also be interesting to estimate the importance of the advection time-lag vs the thermal inertia time-lag.

With $T_c$ we regressed four $T_w$ time series (Basel, Worms, Koblenz and Cologne) along the Rhine. The offset in the this
regression $a_1$, was called RBT, and its change over time was found to be an indicator for anthropogenic HI. The RBT positively correlated to long-term economic changes such as the decrease of nuclear power production as well as to short-term economic events. We showed that changes in production rates (oil refineries or chemical industry) as well as a change in GDP may influence the RBT and therefore the Rhine water temperature. Additionally, the Spearman's Rank correlation between RBT and GDP is positive and significant, delivering another indication for the relation. This case study might deliver a tool for
better understanding of long-term consequences of industrial water use and it might be used as a verification tool for reported

HI. Germany has a rigorous reporting system on cooling water use. However, other countries could check if industrial HI is in accordance with legislative guidelines, without depending on official reports. Whether the ongoing COVID-19 (2020) pandemic and its impact on the economy is also visible using the offered procedures, will need to be proven after the crisis. Hardenbicker et al. (2016) estimated, using a physical model (QSim), that between the reference period of 1961-1990 and the near future 2021-2050 the mean annual $T_w$ of the Rhine could increase by 0.6 $^oC$-1.4 $^oC$. This trend is plausible, according to the historical data analyzed, if the $T_a$ increase remains constant. However, they used a constant anthropogenic HI by e.g. power plants and production industries and different warming rates along the Rhine can result from changes in anthropogenic HI. Next to the global air temperature increase, the industrial use of river water is advised for the future Rhine water temperature. The difference of the $T_w$ warming rate between Basel and the other monitoring stations in the time-series data can be explained by the change in nuclear power production and the influence of general industrial production. This calls for a more integrative river water management than today. For the river Rhine a decreasing (except for Basel) RBT which indicates a decreasing HI, was found. Other river catchment areas with growing energy intensive industries might be impacted by much larger warming rates than those caused by the general increase of $T_a$, experiencing all consequences for physical, chemical and biological processes.

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

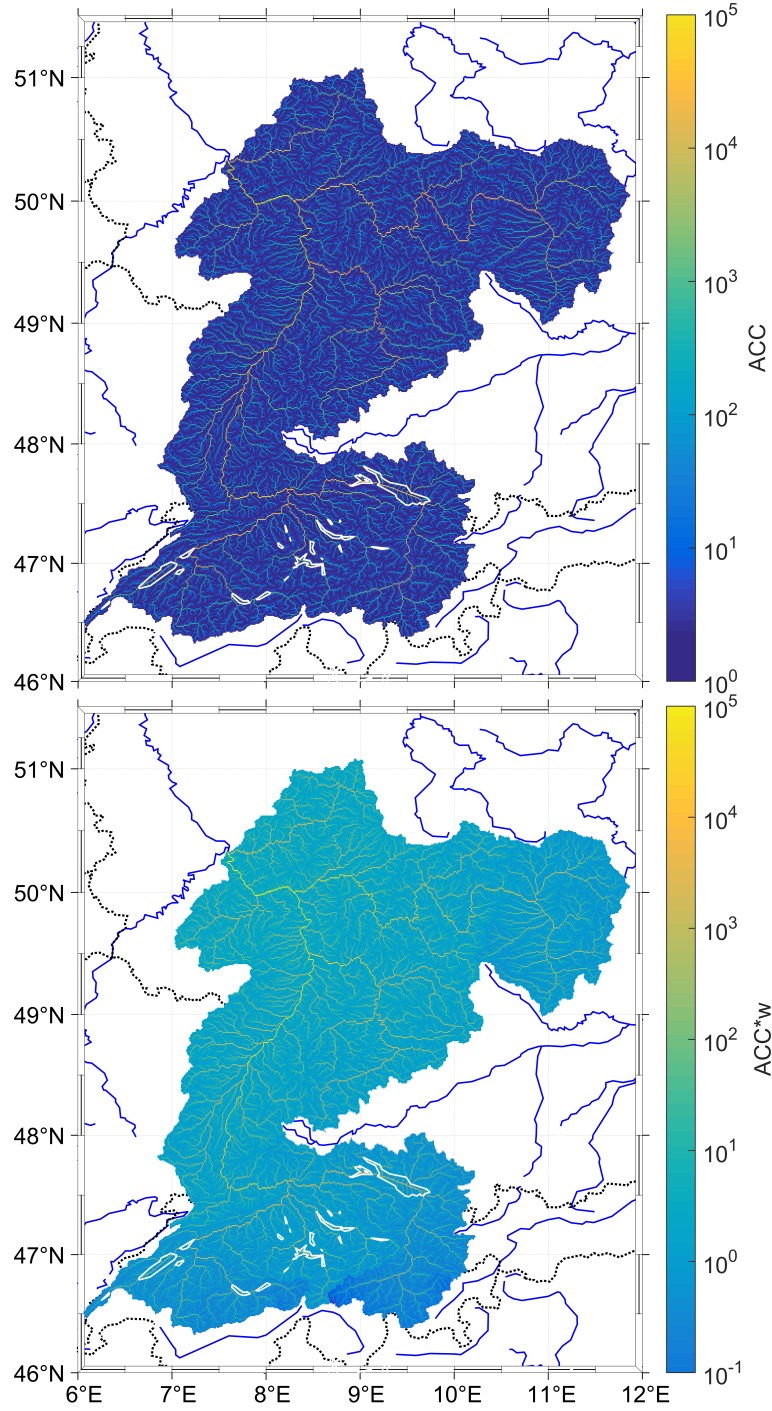

**Figure 3.** Both panels show the catchment area of the Koblenz monitoring station. Top: Number of grid points $ACC$ flowing into each specific grid point. Bottom: $ACC \cdot w$, distance and $ACC$ weighed grid cells.

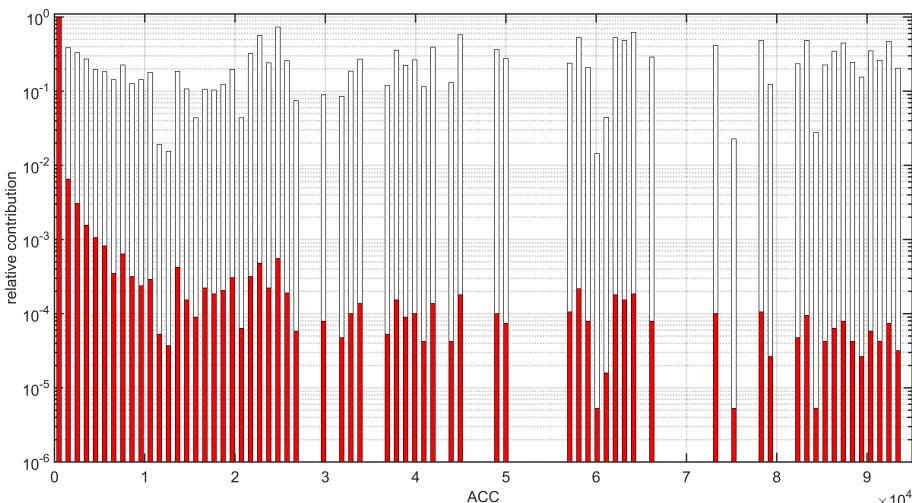

**Figure 4.** $ACC$ bins (x-axis) vs the relative contribution (y-axis). The grid points are binned by their $ACC$ value. The red bars show the relative contribution (largest contribution normalized to one) by the number of grid points in this bin only. The white bars show the distribution using the number of grid points in this bin and weighing $ACC \cdot w$.

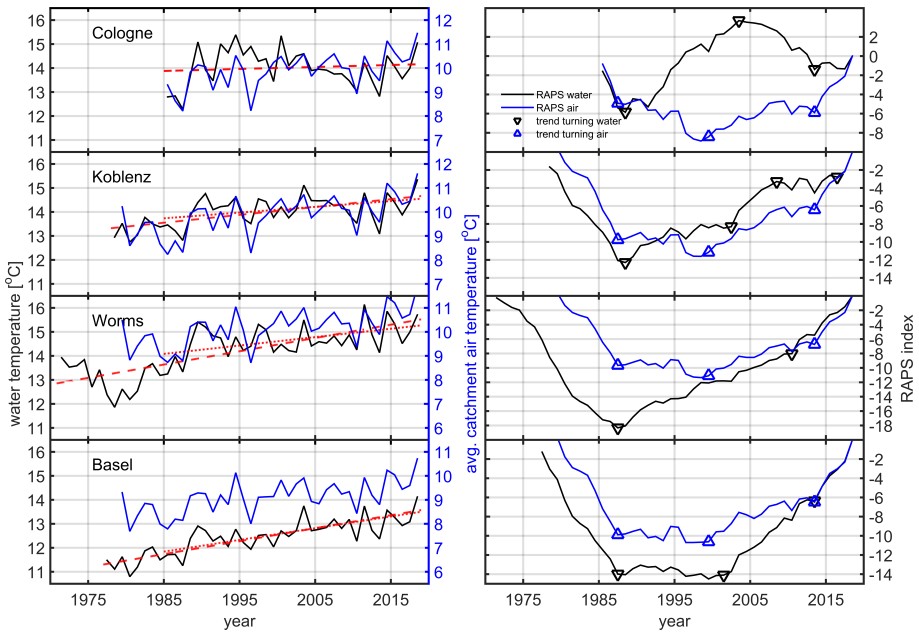

**Figure 5.** Left column: Yearly averages of water temperatures at the four monitoring stations (black line). The red-dashed line is a fit to the available data-set. The red-dotted line is a fit to the overlapping time period (1985-2018). The blue line is the yearly average air temperature of the catchment area.

Right Column: RAPS $T_w$ (black) and $T_a$ (blue) indexes. The triangle markers divide the RAPS index into sections based on a slope change in the RAPS index. Each section also represent a trend-change in the original $T_a$ and $T_w$ time-series.

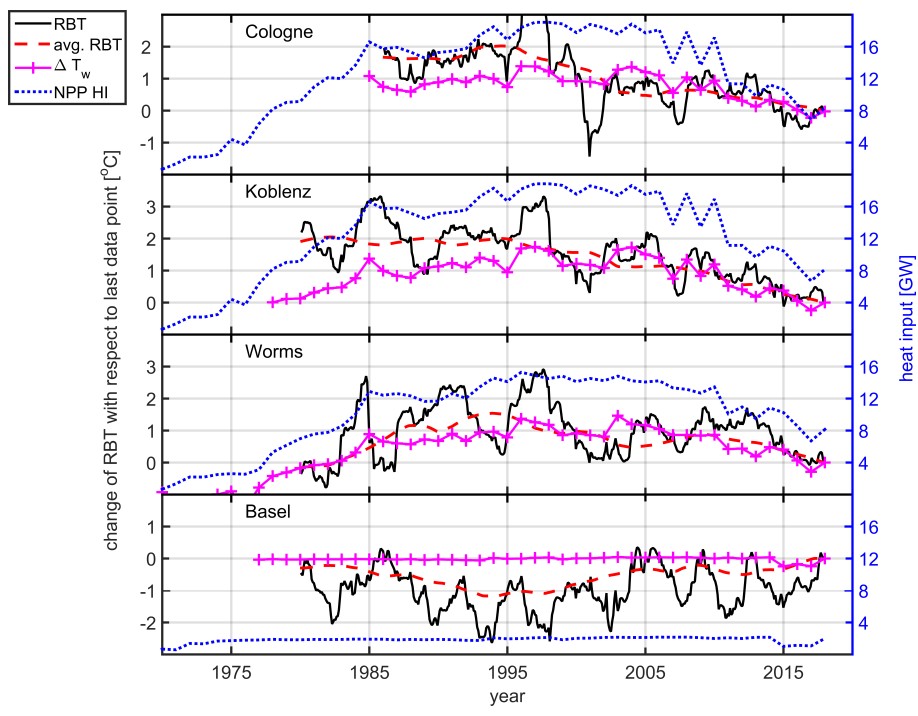

**Figure 6.** RBT from four monitoring stations (black solid line). The red dashed line is the RBT four-year running mean. The magenta line with the + markers shows the ΔRBT relative to the last year. The blue dotted line is the upstream HI by NPPs, Sec. (2.3).

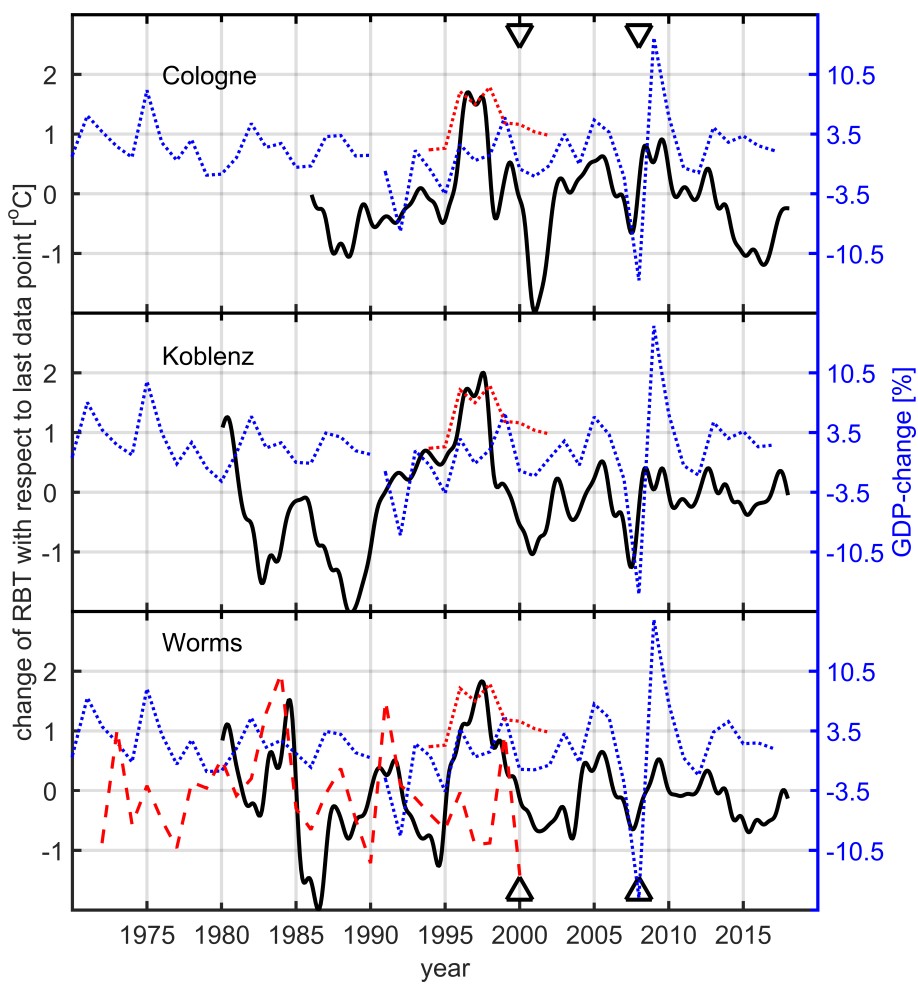

**Figure 7.** The change of RBT (black solid line) at three monitoring stations (Colgone, Koblenz, Worms). The blue dashed line is the GDP-change of the adjacent federal states. To explain trends during two time periods the red dashed line, which is the turnover of the BASF company, and the red dotted line, production rate of the oil refineries, were added. The triangles mark the years 2000 (burst of the dot-com bubble) and 2008 (mortgage crisis).

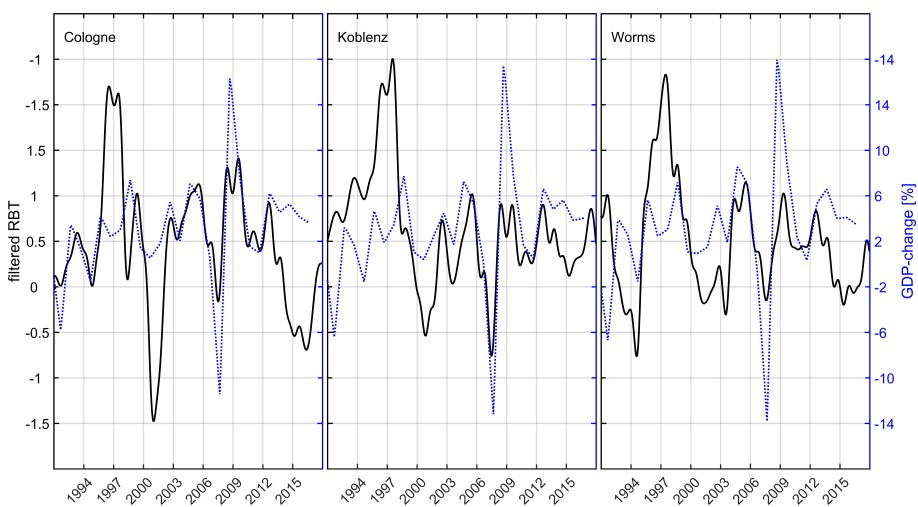

**Figure 8.** Detrended and filtered RBT signal (black solid) and the GDP change (blue dashed) at Cologne, Koblenz and Worms.