# Peer review of "Anthropogenic Influence on the Rhine water temperatures"

_Hydrology and Earth System Sciences, 2019_

## Referee Comment (RC1) · Anonymous Referee #1 · 26 Nov 2019

**GENERAL COMMENTS**

The manuscript presents a study of short term and long term changes in river temperature and investigates the influence of natural and anthropogenic drivers of these changes which is interesting and generally within the scope of HESS. River temperatures at various monitoring locations along the river Rhine as well as industrial production and nuclear power plant activities are analysed. The authors further develop a novel approach of calculating a catchment-wide average air temperature which is used in the linear regression relationship between air and river temperature.

Overall, the scientific approach and the methods appear to be valid. However, there are some points which need further clarification:

[Figure]

(1) The relationships between river temperature and its drivers are investigated using multiple linear regressions separating the so-called Rhine base temperature (i.e. the river temperature without influences of air temperature and discharge) and air temperature and discharge influences on river temperature. More information on the multiple linear regressions for each location is required for the reader to be able to evaluate the robustness of this approach

(2) The computation of the catchment-wide average air temperature is based on the air temperature in each grid cell of the catchment area and the hydrological distance to the river temperature monitoring station assuming a constant flow speed. It would be interesting on what basis the constant flow speed has been derived and how the flow speed varies in space and time and what is the justification of combining a rather complex averaging method of air temperature with a constant flow speed. In order to show the benefits of this rather complex method, benchmarking with simple approaches (e.g. catchment average air temperature in combination with constant lag time, as in Pohle et al., 2019) is suggested.

(3) A data filter is used to compare river temperature and gross domestic product. It would be interesting how the filter parameters have been chosen and how sensitive the results are to different values of these filter parameters.

(4) As short-term and long-term changes of river temperature and its drivers are presented, it would be interesting to know if the data also show statistically significant trends and change points.

The introduction section would benefit from more information and references to recently published literature. Also, the results need to be discussed with reference to related work and including appropriate reference to studies on river temperature. To that end, the authors are suggested to further familiarize with recently published studies on factors influencing river temperature (e.g. Garner et al., 2017; Lisi et al., 2015), river temperature modelling (e.g. Ketabchy et al., 2019; Wondzell et al., 2019; Zhu

et al., 2019) as well as short-term and long-term changes in river temperature and its drivers (e.g. Basarin et al., 2016; Caldwell et al., 2015; Isaak et al., 2018; Pohle et al., 2019).

The manuscript is overall well-written and structured. The results section includes many statements which would be better suited in the methods section. Further, I suggest adding a separate discussion section.

SPECIFIC COMMENTS

Page 1 – line 22 Probably it is better to use "physical based" than "physical". Also, please check whether "deterministically" is the right term – probably it is referred to statistical models?

Page 2 – line 6/7 Is the statement by Markovic true for all rivers? (Their paper refers to Elbe & Danube.)

Page 2 – line 20 The equation is very specific and may be better suited in the "methods" part.

Page 2 – line 21 Suggestion to define coefficents already directly below the equation.

Page 2 – line 25/26 Is this statement universal or only valid for the rivers studied in the cited papers – in that case please name these rivers.

Page 3 – line 3 What is the original temporal resolution of the datasets? What were the procedures for quality control and have there been missing values?

Page 4 – Fig. 1 Please revise the map: make the river Rhine more visible, include monitoring stations and NPPs. Do the time lags refer to hydrological distance or to the grid? How have 0.733 m/s been derived? How robust is this number – I would assume spatial & temporal variability of flow speed.

Page 5 – tab. 2 How exactly have these values been derived?

Page 7 – line 10 Sentence not needed.

Page 7 – line 13-15 Suggest moving sentence to "methods" section.

Page 8 – Fig. 3, tab. 3 Suggest adding 2nd figure column for air temperature. Merge figure and table (i.e. add slope values to the table). Please check robustness of number of digits of slopes, also state whether slopes are statistically significant.

Page 8 – line 3 Which difference? It is stated that Ta warming rates are not really different.

Page 9 – line 3/4 Please be more specific what is meant with "average European river"

Page 9 – line 9/10 Move to "methods" section.

Page 9 & 10 Combine tab. 4 & 5 and highlight the best model for each criterion & location

Page 11 - tab. 6 What does "GW" stand for? Omit "the table shows"

Page 11 – line 16 What is meant with "on average constant" – what time step does the average refer to?

Page 11 – line 25 Why has this particular company (BASF) been chosen?

Page 12 – line 2 Provide test statistics for significance or reword.

Page 14 – line 2 Linear models have also been applied elsewhere. However, it is unclear from this sentence how a linear relationship between air and river temperature implies universal applicability of the method presented in this paper. Furthermore, Morrill et al. found a better fit of non-linear models which might be even more pronounced outside of the tropics (i.e. conditions when air temperature, unlike river temperature, goes far below 0°C)

Page 15 – line 8 For reproducibility, please also name the data providers.

TECHNICAL CORRECTIONS

Page 1 – line 2/3 Sentence unclear – please revise.

Page 1 – line 15 What does "their" refer to?

Page 2 – line 8 Please revise sentence structure.

Page 2 – line 16 Please correct spelling to "assess"

Page 2 – line 24 Is the Markovic reference at the correct position of the sentence?

Page 3 – tab. 1 Move table into methods section.

Page 3 – line 13 Please correct to "European Centre for Medium-Range Weather Forecast".

Page 3 – line 23 Hydrological distance between what? Noun missing.

Page 4 – line 12 Please consider moving reference to end of sentence.

Page 6 – line 3 "2019" instead of "20019"

Page 7 – line 2 Better "reunification" as "unification" refers to 1871.

Page 9 – line 9&10 Nash-Sutcliffe ("e" missing").

Page 12 – Fig. 5 Y-Axis missing for Worms.

Page 12 – line 3 Remove duplicate "by a".

Page 13 – line 2 Check spelling of "Mineralölraffinerie" and use the official name "Oberrhein" instead of "Karlsruhe".

Page 13 – line 2 Use Author (Year) citation format.

Page 14 – line 2 Remove given name from reference.

Page 14 – line 10 Sentence unclear – "and" missing?

Page 14 – line 15 Use Author (Year) citation format. Suggest to use "physical-based"

rather than "physical"

Page 15 – line 21 Please revise word order of sentence.

Page 15 – line 9 Verb missing?

REFERENCES

Basarin, B., Lukić, T., Pavić, D., & Wilby, R. L. (2016). Trends and multi-annual variability of water temperatures in the river Danube, Serbia. Hydrological Processes, 30(18), 3315–3329. https://doi.org/10.1002/hyp.10863

Caldwell, P., Segura, C., Laird, S. G., Sun, G., McNulty, S. G., Sandercock, M., et al. (2015). Short-term stream water temperature observations permit rapid assessment of potential climate change impacts. Hydrological Processes, 29, 2196–2211. https://doi.org/10.1002/hyp.10358

Garner, G., Malcolm, I. A., Sadler, J. P., & Hannah, D. M. (2017). The role of riparian vegetation density, channel orientation and water velocity in determining river temperature dynamics. Journal of Hydrology, 553(September), 471–485. https://doi.org/10.1016/j.jhydrol.2017.03.024

Isaak, D. J., Luce, C. H., Horan, D. L., Chandler, G. L., Wollrab, S. P., & Nagel, D. E. (2018). Global Warming of Salmon and Trout Rivers in the Northwestern U.S.: Road to Ruin or Path Through Purgatory? Transactions of the American Fisheries Society. https://doi.org/10.1002/tafs.10059

Jackson, F. L., Fryer, R. J., Hannah, D. M., Millar, C. P., & Malcolm, I. A. (2018). A spatio-temporal statistical model of maximum daily river temperatures to inform the management of Scotland's Atlantic salmon rivers under climate change. Science of the Total Environment, 612(January), 1543–1558. https://doi.org/10.1016/j.scitotenv.2017.09.010

Ketabchy, M., Sample, D. J., Wynn-Thompson, T., & Yazdi, M. N. (2019).

Simulation of watershed-scale practices for mitigating stream thermal pollution due to urbanization. Science of the Total Environment, 671, 215–231. https://doi.org/10.1016/j.scitotenv.2019.03.248

Lisi, P. J., Schindler, D. E., Cline, T. J., Scheuerell, M. D., & Walsh, P. B. (2015). Watershed geomorphology and snowmelt control stream thermal sensitivity to air temperature. Geophysical Research Letters, 42(9), 1–9. https://doi.org/10.1002/2015GL064083.Received

Piccolroaz, S., Calamita, E., Majone, B., Gallice, A., Siviglia, A., & Toffolon, M. (2016). Prediction of river water temperature: a comparison between a new family of hybrid models and statistical approaches. Hydrological Processes, 30(21), 3901–3917. https://doi.org/10.1002/hyp.10913

Pohle, I., Helliwell, R., Aube, C., Gibbs, S., Spencer, M., & Spezia, L. (2019). Citizen science evidence from the past century shows that Scottish rivers are warming. Science of the Total Environment, 659, 53–65. https://doi.org/10.1016/j.scitotenv.2018.12.325

Wondzell, S. M., Diabat, M., & Haggerty, R. (2019). What Matters Most: Are Future Stream Temperatures More Sensitive to Changing Air Temperatures, Discharge, or Riparian Vegetation? Journal of the American Water Resources Association, 55(1), 116–132. https://doi.org/10.1111/1752-1688.12707

Zhu, S., Heddam, S., Nyarko, E. K., Hadzima-Nyarko, M., Piccolroaz, S., & Wu, S. (2019). Modeling daily water temperature for rivers: comparison between adaptive neuro-fuzzy inference systems and artificial neural networks models. Environmental Science and Pollution Research, 26(1), 402–420. https://doi.org/10.1007/s11356-018-3650-2

---

## Referee Comment (RC2) · Anonymous Referee #2 · 1 Dec 2019

**General comments**

In this study, the authors analyze the effects of Nuclear Power Plants on river water temperature of the Rhine. The authors propose a multiple linear regression model where river water temperature is simulated based on air temperature and streamflow as predictor variables. Air temperature is evaluated through an averaging procedure that accounts for the geomorphology of the hydrological catchment. The intercept of the multiple linear regression model is used as a proxy for the anthropogenic impact on river water temperature and is compared to the time series of GDP and heat input from NPPs.

The presentation of the methodological approach and of the results should be improved, both in terms of clarity and quality. In my opinion the robustness of some

methodological aspects is weak (e.g., the use of a constant flow velocity, the interpretation of the multiple linear regression intercept as "indicator for industrial heat input") and the discussion of the results should be expanded and deepened. The literature review on modeling of river water temperature and assessment of anthropogenic impacts should be updated and the grammar and syntax of the manuscript should be checked carefully. Please, find below some specific comments.

**Specific comments**

Introduction:
The literature review on modeling of river water temperature should be expanded and updated including the most recent studies in this field. Besides "classical" deterministic and statistical models, there is a wide range of models based on machine learning techniques or hybrid physically-based/statistical approaches (e.g., Sahoo et al., 2009; Toffolon and Piccolroaz, 2015; Sohrabi et al., 2017), which have been emerging in the last years. Despite it is not recent, I suggest giving a look to the review paper by Benyahya et al. (2007), which provides a good overview of deterministic and statistical models used in the field of river water temperature prediction. Another useful and more recent paper is that by Gallice et al. (2015). In addition, the authors should refer also to existing literature on the assessment of anthropogenic impact on river water temperature (e.g., Cai et al., 2018; Gaudard et al., 2018; Raman Vinna et al., 2018, just to cite some recent papers).

In general, I believe that the paragraph from P1, line 19 to P2, line 8, should be thoroughly restructured and revised, and the authors should be more precise throughout the text (e.g., at P1, line 22: I believe that the authors intend deterministic and statistical models here; at P2, lines 21-23, the sentence is unclear; at P2, lines 25-26, the comment is superfluous since in a multiple linear regression, such as the one used by the authors, these components are obviously neglected).

P2, lines 7-8: I would rephrase this sentence in more general terms, because the

amount of variance in river water temperature explained by air temperature and stream-flow are strongly dependent on the case study (hydrological regime, season, etc.). In this regard, the authors should expand the analysis of parameters a2 and a3 of their regression model.

The second half of the Introduction (from P2, line 16) should be moved to the methods section and should be improved, as in its current form it does not clearly describe how the authors set up their analysis, especially concerning the definition and use of RBT as an "indicator for industrial heat input" and the time resolution of the data used in the multiple linear regression analysis.

Figure 1
This figure should be updated with the location of the monitoring station and of the NPPs. The main course of the Rhine should also be indicated.

Section 2.1.
I agree on the comment about accuracy and precision, however I wonder if the measurements are affected by instrumental drift and, in case, if the dataset has been corrected accordingly. P3, line 9: this sentence is unclear. In general, I agree that water temperature is rather homogeneous at a river section if it has a compact geometry, while it may be non-uniform if the geometry is complex.

Section 2.2.
Here the authors used a constant flow speed to evaluate the flow time required to travel from a cell of the catchment to the catchment outlet. The authors should clarify how they selected this flow speed and if it is reasonable to assume a constant value (was this velocity the same for the four outlets?). I wonder about the methodological robustness of the approach proposed by the authors since they applied the same flow velocity to all cells pertaining to the catchment, thus both to hillslope and river network cells. In this regard, I also do not fully agree on the sentence at P5, lines 21-22 since before reaching the channel network, rainfall may follow different paths (infiltration,

runoff, etc.), thus exchanging heat with the surrounding environment and decreasing its correlation to Ta. P3, line 20 and P4, line 1: this sentence is unclear.

Section 2.3

The authors state that parameter a1 (the intercept) summarizes all effects that are not directly ascribable to Ta and Q, which "are mostly from anthropogenic sources". Personally, I do not agree that, in general, the value of a1 can be unequivocally related to anthropogenic factors. The authors should support this statement referring to previous literature on the topic. In this regard, a useful reading is Isaak et al (2011), where also the multiplicative interaction term has been included in the multiple linear regression model. Variables $x_0$, $y_0$, and $t_t$ in eq 2 are not defined.

Table 2 (and corresponding description in the main text): the authors should provide details on why they assumed a linearly deceasing weighting factor instead of other weighting functions. While the weighting factors decreases with $\Delta t$, I expect that $T_w$ is no more correlated to $T_a$ after some time. The authors obtain the best results using the "Time lag" model instead of the "Time lag + weight" model, saying that the furthest and oldest $T_a$ influences on $T_w$ are still carried as information in the water mass (P9, lines 4-5). In my opinion, the real reason is that without assuming a deceasing weighting factor the authors increase the dependence of current river water temperature on previous conditions, thus implicitly accounting for the thermal inertia of the river. This is an important aspect controlling river water temperature, which is not explicitly included in the model proposed by the authors and that can be accounted for e.g., through autocorrelation terms (e.g., Caissie et al., 2001; Toffolon and Piccolroaz, 2015).

Control scenarios

I would use a different word than "scenarios" here, since these are not scenarios but different approaches to calculate $T_c$.

Section 2.4

The authors should explain how they calculated the heat input by NPP to the Rhine.

The section should be expanded, and the sentences harmonized to make the reading more fluid (too short sentences).

Figure 3 and Table 3
Figure 3 would benefit from the inclusion of the air temperature time series with the corresponding linear trends. This would be useful for better understanding the correlation between river water temperature and air temperature fluctuations, which are filtered out when using linear trends. In this regard, it would be useful to add the Pearson correlation coefficient between these two variables in Table 3. At P8, lines 12-15 it would be useful to compare the trends found by the authors with those of more recent studies.

Tables 4 and 5
Why did the authors use the "Time lag+weight" approach for all other results instead of the "Time lag" approach, which performed the best? It should be clearly indicated if the RMSE and NSC refer to daily or annual values.

Section 3.3
It is unclear how the authors evaluated RBT over time. Did I correctly understand that they applied the multiple linear regression model for overlapping two-year time windows shifted by one month? What was the rationale of assuming two-year time windows instead of longer periods? Are the results affected by the length of the time window used for this analysis? P10, line 2 and P11, line 4: these sentences are qualitative, and not sufficiently supported by the results. The comment on the effect of alpine lakes is not well connected to the rest of the paragraph and should be expanded with some more detailed discussion. Eq 10 is dimensionally not consistent. How did the authors select the periods in Table 6? The authors could do the same calculation in continuous, for the entire period when the data are available (e.g., using the same two-year time windows as before). P11, line 16: what is the BASF company? This should be explained. Why RBT in Figures 4 and 5 are different? How sensitive are the results of the correlation analysis to the filtering of the data? How the filtering parameters have

been chosen and why 480 days has been used to shift the GDP-change time series? This number seems quite arbitrary.

Appendices

Appendices could be moved to the main text. In particular, the sentences in Appendix B should be revised because they have some syntax errors and typos. Figures A1 and A2 are inverted and the caption is the same. The analysis of parameters $a_2$ and $a_3$ should be deepened and moved to the main text.

**Technical corrections**

P1, line 13: "but an" –> "but is an". Is "means of production" an appropriate term in this context?

P2, line 3 and following lines: the use of "Ta –> Tw" is informal and should be modified.

P2, line 8: "hydro-logical" –> "hydrological"

P2, lines 8-9: a reference is needed here.

P2, line 16: is "revise" the most appropriate term here?

P2, line 20: "almost ideal" –> "ideal", "interesting", "meaningful"

P4, line 13: "followed, by" –> "followed by". Please, thoroughly revise the punctuation throughout the article (use of commas, missing close-brackets, etc).

P5, line 17: "ptovided" –> "provided"

P6, line 1: I would say that authors present four $T_c$ calculations, not two.

P6, line 18: "heat input by NPPsto the Rhine" –> "heat input by NPP to the Rhine"

P8, line 5: "(0.0350 $^\circ Cy^{-1}$)" –> "(0.0489 $^\circ Cy^{-1}$)"

P10, line 15: "over the a time period" –> "over a time period"

P11, line 1: "shorter timer scale but do not seem,to our" –> "shorter time scale but do

not seem, to our"

P11, line 14: "A a discontinuity" –> "A discontinuity"

P11, line 19: "by a by a" –> "by a"

**References**

Benyahya L., Caissie D., St.Hilaire A., Ouarda T.B.M.J., Bobe B. 2007. A review of statistical water temperature models. Canadian Water Resources Journal 32: 179–192

Cai H., Piccolroaz S., Huang J., et al. 2018. Quantifying the impact of the Three Gorges Dam on the thermal dynamics of the Yangtze River. Environ Res Lett.;13(2018):054016.

Caissie D., El Jabi N., Satish M.G. 2001. Modelling of maximum daily water temperatures in a small stream using air temperature. Journal of Hydrology 251: 14–28.

Caissie D., Satish M.G., El-Jabi N. 2005. Predicting river water temperatures using the equilibrium temperature concept with application on Miramichi River catchments (New Brunswick, Canada) Hydrol. Process. 19 2137–59

Gaudard, A, Weber, C, Alexander, TJ, Hunziker, S, Schmid, M. 2018. Impacts of using lakes and rivers for extraction and disposal of heat. WIREs Water. 5:e1295.

Gallice A., Schaefli B., Lehning M., Parlange M.B., and Huwald H. 2015. Stream temperature prediction in ungauged basins: review of recent approaches and description of a newphysics-derived statisticalmodel. Hydrol Earth Syst Sci 19:3727-3753

Isaak D.J., Luce C.H., Rieman B.E., Nagel D.E., Peterson E.E., Horan D.L., Parkes S., Chandler, G.L. 2010. Effects of climate change and wildfire on stream temperatures and salmonid thermal habitat in a mountain river network. Ecol. Appl. 20, 1350–1371.

Isaak D.J., Wollrab S., Horan, D., Chandler, G. 2011. Climate change effects on stream

and river temperatures across the northwest U.S. from 1980-2009 and implications for salmonid fishes. Climatic Change. 113: 499-524.

Sahoo G.B., Schladow S.G., and Reuter J.E. 2009. Forecasting stream water temperature using regression analysis, artificial neural network, and chaotic non-linear dynamic models. J. Hydrol. 378 325–42

Sohrabi M.M., Benjankar R., Tonina D., Wenger S.J., and Isaak D.J. 2017. Estimation of daily stream water temperatures with a Bayesian regression approach. Hydrol. Process. 31, 1719–1733

Toffolon M., and Piccolroaz S. 2015. A hybrid model for river water temperature as a function of air temperature and discharge. Environmental Research Letters 10: 114011.

van Vliet M.T.H., Ludwig F., Zwolsman J.J.G., Weedon G.P., Kabat P. 2011. Global river temperatures and sensitivity to atmospheric warming and changes in river flow. Water Resour. Res. 2011, 47

Raman Vinna L., Wüest A., Zappa M., Fink G., Bouffard, D. 2018. Tributaries affect the thermal response of lakes to climate change, Hydrol. Earth Syst. Sci., 22, 31–51

---

## Author Comment (AC1) · 13 Jan 2020

Interactive comment on "Anthropogenic Influence on the Rhine water temperatures "by Alex Zavarsky and Lars Duester

Anonymous Referee #1

Introduction by the Authors: We would like to sincerely thank both reviewers for the comments and thoughts about our work and this manuscript. We think that the input significantly improved the manuscript. Based on the reviewers comments and by reviewing the code once again transposed numbers were found in the coding. By correcting the calculation method ACC*w provides the lowest RMSE and largest NCS in three out of four station. At the same time we were able to further decrease the RMSE

for the ACC*w calculation method. The reasons for the ACC*w resulting in lower RMSE compared to ACC or w only, is now described in detail in the methods section. Overall, the results (correlations, RMSE, NCS, ΔRBTcalc) changed only so slightly, that the scientific conclusion and the key messages were not influenced. This also visible in the attached track changes version of the manuscript.

GENERAL COMMENTS

The manuscript presents a study of short term and long term changes in river temperature and investigates the influence of natural and anthropogenic drivers of these changes which is interesting and generally within the scope of HESS. River temperatures at various monitoring locations along the river Rhine as well as industrial production and nuclear power plant activities are analyzed. The authors further develop a novel approach of calculating a catchment-wide average air temperature which is used in the linear regression relationship between air and river temperature. Overall, the scientific approach and the methods appear to be valid. However, there are some points which need further clarification:

(1) The relationships between river temperature and its drivers are investigated using multiple linear regressions separating the so-called Rhine base temperature (i.e. the river temperature without influences of air temperature and discharge) and air temperature and discharge influences on river temperature. More information on the multiple linear regressions for each location is required for the reader to be able to evaluate the robustness of this approach

Comment: The RMSE and the NCS information is provided for every measurement station. In addition, the data is now included in the supplement. We used the year 2001 as a test year and regressed Tw using Tc and Q just for this year. Then the 2001 regression coefficients were used to calculate a modelled Tw for the years 2000 to 2018. The RMSE and NCS show better results compared to the long term regression, which is sensible for a shorter regression period. The RMSE and NCS for each year

from 2000 to 2018 follow the same pattern among the calculation methods. This means that the ACC*w method is always the best at three stations and the methods without time lag always show a larger RMSE than the ones with time lag (2) The computation of the catchment-wide average air temperature is based on the air temperature in each grid cell of the catchment area and the hydrological distance to the river temperature monitoring station assuming a constant flow speed. It would be interesting on what basis the constant flow speed has been derived and how the flow speed varies in space and time and what is the justification of combining a rather complex averaging method of air temperature with a constant flow speed. In order to show the benefits of this rather complex method, benchmarking with simple approaches (e.g. Catchment average air temperature in combination with constant lag time, as in Pohle et al., 2019) is suggested.

Comment: In our model, the flow speed does not vary in space and time. Generally, the flow speed in the shipping channel is between 1 m/s and 2 m/s. This is supported by ADCP round robin tests (https://www.bafg.de/DE/05_Wissen/02_Veranst/2007/10-09-07_bericht.pdf?__blob=publicationFile) which showed a average flow speed of 1.2 m/s. Using the Koblenz data as reference we tried several flow speeds to minimize the RMSE of the model. We found a minimum of RMSE at 0.4 m/s. This is in the extended-range flow speeds. We expected a higher correlation at lower flow speeds than actually measured in the Rhine as we do not model standing water bodies. To us a flow speed with a magnitude difference would be questionable, but the one used is within reasonable limits.

(3) A data filter is used to compare river temperature and gross domestic product. It would be interesting how the filter parameters have been chosen and how sensitive the results are to different values of these filter parameters.

Comment: We used a Butterworth band-pass filter instead of a running mean filter because the filter function of a butterworth is much easier to understand and it simply cuts all variations that are outside of the pass area. In this manuscript everything with

a periodicity of 20 years (0.05 y-1) or longer is cut off. The reason is to eliminate long term trends, because the aim is to compare RBT to the GDP change. The lower limit is 0.9 years (1.1 y-1). Fast variations (faster than a year) of the RBT could influence the correlation vs a data-set (here the GDP) which is provided on a yearly basis. Therefore we smoothing is needed.

(4) As short-term and long-term changes of river temperature and its drivers are presented, it would be interesting to know if the data also show statistically significant trends and change points. The introduction section would benefit from more information and references to recently published literature. Also, the results need to be discussed with reference to related work and including appropriate reference to studies on river temperature. To that end, the authors are suggested to further familiarize with recently published studies on factors influencing river temperature (e.g. Garner et al., 2017; Lisi et al., 2015), river temperature modelling (e.g. Ketabchy et al., 2019; Wondzell et al., 2019; Zhu et al., 2019) as well as short-term and long-term changes in river temperature and its drivers (e.g. Basarin et al., 2016; Caldwell et al., 2015; Isaak et al., 2018; Pohle et al.,2019).The manuscript is overall well-written and structured. The results section includes many statements which would be better suited in the methods section. Further, I suggest adding a separate discussion section.

Comment: Thank you for pointing out additional literature. We added the rescaled adjusted partial sums to the manuscript. We checked trends of Tw and Ta at the four measurement stations and differences are visible. These differences are in accordance with our hypothesis that the progress of Tw at Worms, Koblenz and Mainz cannot be fully explained by the trend of Ta.

SPECIFIC COMMENTS

Page 1 – line 22 probably it is better to use "physical based" than "physical". Also, please check whether "deterministically" is the right term – probably it is referred to statistical models?

[Figure]

Comment: Thank you, we changed the wording.

Page 2 – line 6/7 Is the statement by Markovic true for all rivers? (Their paper refers to Elbe & Danube.)

Comment: We added the information that their study is based on Elbe and Danube data. As these two rivers are more or less comparable in size and catchment area to the Rhine, we think and also show that consistent results are given.

Page 2 – line 20 The equation is very specific and may be better suited in the "methods "part.

Comment: Thank you for the comment, but the fundamental idea of our hypothesis is to use the regression coefficients as explanation for changes in Tw. Therefore we need a simple linear Ta→Tw model. We want to present this idea and thought process in the Introduction. This is also done because we want to explain why we do not use hybrid, exponential models.

Page 2 – line 21 Suggestion to define coefficients already directly below the equation.

Comment: Thank you, we changed it.

Page 2 – line 25/26 Is this statement universal or only valid for the rivers studied in the cited papers – in that case please name these rivers.

Comment: We reorganized the references and specified to which subject the references addressed.

Page 3 – line 3 what is the original temporal resolution of the datasets? What were the procedures for quality control and have there been missing values?

Comment: The original resolution is 10 min. We added a line to missing values and resolution in Sec. 2.1. The quality control is done by the sources. They initially verify the data-set. Additionally, the data-set was screened by us for suspicious features.

Page 4 – Fig. 1 Please revise the map: make the river Rhine more visible, include monitoring stations and NPPs. Do the time lags refer to hydrological distance or to the grid? How have 0.733 m/s been derived? How robust is this number – I would assume spatial & temporal variability of flow speed.

Comment: We revised Fig. 1 which is now Fig. 2. The NPPs and measurement stations are now also included. We also describe in Sec. 2.7 how we obtained the flow speed and compare it to measured flow speeds.

Page 5 – tab. 2 How exactly have these values been derived?

Comment: We changed the table caption and added a few sentences in the "weighing coefficients" subsection, answering the question.

Page 7 – line 10 Sentence not needed.

Comment: We removed this sentence.

Page 7 – line 13-15 Suggest moving sentence to "methods" section.

Comment: These lines briefly explain Fig. 3. Hence, the authors think it should better remain in the Results section.

Page 8 – Fig. 3, tab. 3 Suggest adding 2nd figure column for air temperature. Merge figure and table (i.e. add slope values to the table). Please check robustness of number of digits of slopes, also state whether slopes are statistically significant.

Comment: We reduced the number of digits and added $R^2$ values and a significance statement. We also added Ta in the figure and the RAPS index for trend analysis.

Page 8 – line 3 Which difference? It is stated that Ta warming rates are not really different. Comments: We added "from each other" to clarify this sentence.

Page 9 – line 3/4 Please be more specific what is meant with "average European river"

Comment: We removed this part.

Page 9 – line 9/10 Move to "methods" section.

Comment: This is a brief reminder and explanation for Tab. 5. We prefer to keep it there.

Page 9 & 10 Combine tab. 4 & 5 and highlight the best model for each criterion & location

Comment: We combined the tables and highlighted the best model.

Page 11 - tab. 6 what does "GW" stand for? Omit "the table shows"

Comment: Thank you, we replaced GW with "$\Delta$Hi [GW]". We removed "the table shows".

Page 11 – line 16 What is meant with "on average constant" – what time step does the average refer to?

Comment: The sentence was completely revised. P 16 Line 289.

Page 11 – line 25 Why has this particular company (BASF) been chosen?

Comment: It is close to the measurement station Worms and also provides significant heat input. We added this information to the manuscript.

Page 12 – line 2 Provide test statistics for significance or reword.

Comment: We omitted the word significant.

Page 14 – line 2 Linear models have also been applied elsewhere. However, it is unclear from this sentence how a linear relationship between air and river temperature implies universal applicability of the method presented in this paper. Furthermore, Morrill et al. found a better fit of non-linear models which might be even more pronounced outside of the tropics (i.e. conditions when air temperature, unlike river temperature, goes far below 0◦C)

Comment: The scope of this paper is not only finding a better (lower RMSE) way to

model Tw, but to apply coefficients of a linear regression to better explain trend in Tw. more precise (etc.) models might be available, but most of them don't allow to distinct between anthropogenic, meteorological and hydrological impacts. If they allow this distinction, they are very labor-, time-, staff- and computing capacity intensive. This is not the case for the model proposed by us. Morrill et al. found suitable linear relationships between Ta and Tw for rivers around the world. This was a prerequisite for our analysis.

Page 15 – line 8 for reproducibility, please also name the data providers.

Comment: The data providers are mentioned in the methods section.

TECHNICAL CORRECTIONS

Page 1 – line 2/3 Sentence unclear – please revise.

Comment: Changed.

Page 1 – line 15 What does "their" refer to?

Comment: It refers to: energy intensive industries such as power plants, oil refineries, paper or steel mills. Changed to: "Its availability is a basic requirement for the facilitie's location (Förster and Lilliestam, 2010).

Page 2 – line 8 Please revise sentence structure.

Comment: We revised the sentence.

Page 2 – line 16 Please correct spelling to "assess"

Comment: Thanks we changed it.

Page 2 – line 24 Is the Markovic reference at the correct position of the sentence?

Comment: We changed the position.

Page 3 – tab. 1 Move table into methods section.

Comment: It is in the methods section. The final formatting is applied by Copernicus.

Page 3 – line 13 Please correct to "European Centre for Medium-Range Weather Forecast".

Comment: Sorry, an awkward mistake. We changed it.

Page 3 – line 23 Hydrological distance between what? Noun missing.

Comment: Corrected.

Page 4 – line 12 Please consider moving reference to end of sentence.

Comment: We moved them.

Page 6 – line 3 "2019" instead of "20019"

Comment: We corrected it.

Page 7 – line 2 Better "reunification" as "unification" refers to 1871.

Comment: Typo, corrected.

Page 9 – line 9&10 Nash-Sutcliffe ("e" missing").

Comment: We added an e.

Page 12 – Fig. 5 Y-Axis missing for Worms.

Comment: We added the axis.

Page 12 – line 3 Remove duplicate "by a".

Comment: Thanks, corrected.

Page 13 – line 2 Check spelling of "Mineralölraffinerie" and use the official name "Oberrhein" instead of "Karlsruhe".

Comment: We changed that.

Page 13 – line 2 Use Author (Year) citation format.

Comment: That's the formatting prescribed by Copernicus.

Page 14 – line 2 Remove given name from reference.

Comment: Changed.

Page 14 – line 10 Sentence unclear – "and" missing?

Comment: We corrected it.

Page 14 – line 15 Use Author (Year) citation format. Suggest to use "physical-based

Comment: That's the formatting prescribed by Copernicus.

REFERENCES

Basarin, B., Lukic, T., Pavic, D., & Wilby, R. L. (2016). Trends and multi-annual variabil-ity of water temperatures in the river Danube, Serbia. Hydrological Processes, 30(18),3315–3329. https://doi.org/10.1002/hyp.10863

Caldwell, P., Segura, C., Laird, S. G., Sun, G., McNulty, S. G., Sandercock, M., et al. (2015). Short-term stream water temperature observations permit rapid as-sessment of potential climate change impacts. Hydrological Processes, 29, 2196–2211.https://doi.org/10.1002/hyp.10358

Garner, G., Malcolm, I. A., Sadler, J. P., & Hannah, D. M. (2017).The role of riparian vegetation density, channel orientation and water velocity in determin-ing river temperature dynamics. Journal of Hydrology, 553(September), 471–485.https://doi.org/10.1016/j.jhydrol.2017.03.024

Isaak, D. J., Luce, C. H., Horan, D. L., Chandler, G. L., Wollrab, S. P., & Nagel, D. E.(2018). Global Warming of Salmon and Trout Rivers in the Northwestern U.S.: Road to Ruin or Path Through Purgatory? Transactions of the American Fisheries Society. https://doi.org/10.1002/tafs.10059

[Figure]

Jackson, F. L., Fryer, R. J., Hannah, D. M., Millar, C. P., & Malcolm, I. A.(2018).A spatiotemporal statistical model of maximum daily river temperatures to inform the management of Scotland's Atlantic salmon rivers under climate change. Science of the Total Environment, 612(January), 1543–1558.https://doi.org/10.1016/j.scitotenv.2017.09.010

Ketabchy, M., Sample, D. J., Wynn-Thompson, T., & Yazdi, M. N. (2019).C6 Simulation of watershed-scale practices for mitigating stream thermal pollution due to urbanization. Science of the Total Environment, 671, 215–231.https://doi.org/10.1016/j.scitotenv.2019.03.248

Lisi, P. J., Schindler, D. E., Cline, T. J., Scheuerell, M. D., & Walsh,P. B. (2015).Watershed geomorphology and snowmelt control stream thermal sensitivity to air temperature. Geophysical Research Letters, 42(9), 1–9.https://doi.org/10.1002/2015GL064083

Piccolroaz, S., Calamita, E., Majone, B., Gallice, A., Siviglia, A., & Toffolon, M. (2016).Prediction of river water temperature: a comparison between a new family of hybrid models and statistical approaches. Hydrological Processes, 30(21), 3901–3917.https://doi.org/10.1002/hyp.10

Comment:Real good paper that shows the difference between Lake fed, regulated, snow fed and low land rivers.

Pohle, I., Helliwell, R., Aube, C., Gibbs, S., Spencer, M., & Spezia, L.(2019).Citizen science evidence from the past century shows that Scottish rivers are warming. Science of the Total Environment, 659, 53–65.https://doi.org/10.1016/j.scitotenv.2018.12.325

Wondzell, S. M., Diabat, M., & Haggerty, R. (2019). What Matters Most: Are Future Stream Temperatures More Sensitive to Changing Air Temperatures, Discharge, or Riparian Vegetation? Journal of the American Water Resources Association, 55(1),116–132. https://doi.org/10.1111/1752-1688.12707

Zhu, S., Heddam, S., Nyarko, E. K., Hadzima-Nyarko, M., Piccolroaz, S.,

& Wu, S.(2019). Modeling daily water temperature for rivers: comparison between adaptiveneuro-fuzzy inference systems and artificial neural networks models. EnvironmentalScience and Pollution Research, 26(1), 402–420. https://doi.org/10.1007/s11356-018-3650-2

Hoef, J. M. V., Peterson, E., & Theobald, D. (2006). Spatial statistical models that use flow and stream distance. Environmental and Ecological Statistics, 13(4), 449–464. doi:10.1007/s10651-006-0022-8   Reviewer 2
In this study, the authors analyze the effects of Nuclear Power Plants on river water temperature of the Rhine. The authors propose a multiple linear regression model where river water temperature is simulated based on air temperature and streamflow as predictor variables. Air temperature is evaluated through an averaging procedure that accounts for the geomorphology of the hydrological catchment. The intercept of the multiple linear regression models is used as a proxy for the anthropogenic impact on river water temperature and is compared to the time series of GDP and heat input from NPPs. The presentation of the methodological approach and of the results should be improved, both in terms of clarity and quality. In my opinion the robustness of some methodological aspects is weak (e.g., the use of a constant flow velocity, the interpretation of the multiple linear regression intercept as "indicator for industrial heat input") and the discussion of the results should be expanded and deepened. The literature review on modeling of river water temperature and assessment of anthropogenic impacts should be updated and the grammar and syntax of the manuscript should be checked carefully. Please, find below some specific comments.

Specific comments

Introduction:

The literature review on modeling of river water temperature should be expanded and updated including the most recent studies in this field. Besides "classical" deterministic and statistical models, there is a wide range of models based on machine learning techniques or hybrid physically-based/statistical approaches (e.g., Sahoo et al., 2009;Toffolon and Piccolroaz, 2015; Sohrabi et al., 2017), which have been emerging in the last years. Despite it is not recent, I suggest giving a look to the review paper by Benyahya et al. (2007), which provides a good overview of deterministic and statistical models used in the field of river water temperature prediction. Another useful and more recent paper is that by Gallice et al. (2015).

Comment: Another thorough literature search was undertaken and we added among other references, the references proposed by both reviewers. The overview of water temperature models was extended in the introduction.

In addition, the authors should refer also to existing literature on the assessment of anthropogenic impact on river water temperature (e.g., Cai et al., 2018; Gaudard et al., 2018; Raman Vinna et al., 2018,just to cite some recent papers).

Comment: The publications were cross-checked. The input was included in the revision of the manuscript.

In general, I believe that the paragraph from P1, line 19 to P2, line 8, should be thoroughly restructured and revised, and the authors should be more precise throughout the text (e.g., at P1, line 22: I believe that the authors intend deterministic and statistical models here; at P2, lines 21-23, the sentence is unclear; at P2, lines 25-26, the comment is superfluous since in a multiple linear regression, such as the one used by the authors, these components are obviously neglected). P2, lines 7-8: I would rephrase this sentence in more general terms, because the amount of variance in river water temperature explained by air temperature and streamflow are strongly dependent on the case study (hydrological regime, season, etc.).

Comment: Thank you for the comments. We revised the whole introduction. The
changes we made can be seen in the track changes version. P2 lines 25-26: We know that our model does exclude ground heat flux and friction. If the parameters are important they would appear most likely and unfortunately in the regression coefficient a1. However, a1 is the basis of our analysis which should display the anthropogenic heat input We just want to say that we think these heaf fluxes are neglible and do not interfere with our anthropogenic heat input. In this regard, the authors should expand the analysis of parameters a2 and a3 of their regression model. The second half of the Introduction (from P2, line 16) should be moved to the methods section and should be improved, as in its current form it does not clearly describe how the authors set up their analysis, especially concerning the definition and use of RBT as an "indicator for industrial heat input" and the time resolution of the data used in the multiple linear regression analysis. Figure 1 This figure should be updated with the location of the monitoring station and of the NPPs. The main course of the Rhine should also be indicated. Comment: We changed Figure 1. In the introduction we give just a basis overview of our idea which is closely linked to the linear regression model. We moved some parts to the methods section. The detailed calculations are described in the methods section.

Section 2.1. I agree on the comment about accuracy and precision, however I wonder if the measurements are affected by instrumental drift and, in case, if the dataset has been corrected accordingly.

Comment: The data was verified by the data provider(e.g., by recurrent validation measurments, recalibration if needed or cross-validation). The data-set was screened for suspicious features. We stated this in the manuscript.

P3, line 9: this sentence is unclear. In general, I agree that water temperature is rather homogeneous at a river section if it has a compact geometry, while it may be non-uniform if the geometry is complex.

Comment: We know that the measured water temperature, especially in complex river

geometries, is an on-spot in-situ temperature and could be different from a cross-section average Tw. However, a method benefit of this analysis is that only the water temperature differences are needed. If the measured Tw changes and the cross-section Tw does, accordingly.

Section 2.2.Here the authors used a constant flow speed to evaluate the flow time required to travel from a cell of the catchment to the catchment outlet. The authors should clarify how they selected this flow speed and if it is reasonable to assume a constant value (was this velocity the same for the four outlets?). I wonder about the methodological robustness of the approach proposed by the authors since they applied the same flow velocity to all cells pertaining to the catchment, thus both to hillslope and river network cells. In this regard, I also do not fully agree on the sentence at P5, lines 21-22 since before reaching the channel network, rainfall may follow different paths (infiltration,C3runoff, etc.), thus exchanging heat with the surrounding environment and decreasing its correlation to Ta.

Comment: Pls. cf. GENERAL COMMENTS (2) to reviewer 1.

P3, line 20:

Comment: We changed the wording.

P4, line 1

Comment: We changed the wording

Section 2.3 The authors state that parameter a1 (the intercept) summarizes all effects that are not directly ascribable to Ta and Q, which "are mostly from anthropogenic sources". Personally, I do not agree that, in general, the value of a1 can be unequivocally related to anthropogenic factors.

Comment: Of course there is no proven, but this the hypothesis. We are able to strongly support this hypothesis by comparing changes in anthropogenic heat input (nuclear power plants) and short term economic changes to a1 and draw a consistent picture in

the manuscript.

The authors should support this statement referring to previous literature on the topic. In this regard, a useful reading is Isaak et al (2011), where also the multiplicative interaction term has been included in the multiple linear regression model.

Comment: We reviewed all citations, thank you for the hints. If applicable we changed the manuscript. Especially, the different methods for modelling Tw are described now more detailed in the introduction.

Variablesx0,y0, and in eq 2 are not defined. Table 2 (and corresponding description in the main text): the authors should provide details on why they assumed a linearly deceasing weighting factor instead of other weighting functions.

Comment: We added an explanation of x,y. We revised our model and use now ACC*w as weighting factor. The reason for a linear decrease cannot be answered within this manuscript and more research is needed.

While the weighting factors decreases with $\Delta t$, I expect that Tw is no more correlated to Ta after some time. The authors obtain the best results using the "Time lag" model instead of the "Time lag + weight" model, saying that the furthest and oldest Ta influences on Tw are still carried as information in the water mass (P9, lines 4-5). In my opinion, the real reason is that without assuming a deceasing weighting factor the authors increase the dependence of current river water temperature on previous conditions, thus implicitly accounting for the thermal inertia of the river. This is an important aspect controlling river water temperature, which is not explicitly included in the model proposed by the authors and that can be accounted for e.g., through autocorrelation terms (e.g., Caissie et al., 2001; Toffolon and Piccolroaz, 2015).

Comment: We think that the reason for using a weighting factor decreasing is a) to put less weight on the large amount of grid-points with less ACC and b) to put less weight on temperatures with a large $\Delta t$. Autocorrelation is an option but we decided not use it

for this model.

Control scenarios I would use a different word than "scenarios" here, since these are not scenarios but different approaches to calculate Tc .

Comment: Changed.

Section 2.4 The authors should explain how they calculated the heat input by NPP to the Rhine. The section should be expanded, and the sentences harmonized to make the reading more fluid (too short sentences).

Comment: We moved the explanation of the NPP heat input to the methods section and revised it.

Figure 3 and Table 3 Figure 3 would benefit from the inclusion of the air temperature time series with the corresponding linear trends. This would be useful for better understanding the correlation between river water temperature and air temperature fluctuations, which are filtered out when using linear trends. In this regard, it would be useful to add the Pearson correlation coefficient between these two variables in Table 3.

Comment: We added air temperature to the figure. We also added the RAPS index to make trends more visible.

At P8, lines 12-15 it would be useful to compare the trends found by the authors with those of more recent studies.

Comment: We removed this section. The focus of the paper is on providing reasons for the heterogeneous Tw trends in the Rhine river, an urgent matter in regulative river heat evaluation in times of climate change.

Tables 4 and 5 Why did the authors use the "Time lag weight" approach for all other results instead of the "Time lag" approach, which performed the best? It should be clearly indicated if the RMSE and NSC refer to daily or annual values.

Comment: As mentioned before (first page of this document), the data was reanalyzed.

[Figure]

As a consequence the tables and parts of the results were revised. The scientific conclusion was not changed.

Section 3.3 It is unclear how the authors evaluated RBT over time. Did I correctly understand that they applied the multiple linear regression model for overlapping two-year time windows shifted by one month? What was the rationale of assuming two-year time windows instead of longer periods? Are the results affected by the length of the time window used for this analysis?

Comment: Longer time windows would decrease the temporal resolution of the regression. A shorter time window increases the influence by other linear dependent influences. The two years were chosen to address two full annual cycles. If a year was extraordinary concerning air temperature or discharge, a two year cycle would not be prone to such events.

P10, line 2 these sentences are qualitative, and not sufficiently supported by the results.

Comment: We changed the wording. We add that we cannot meaningfully interpret the absolute value RBT.

P11, line 4: these sentences are qualitative, and not sufficiently supported by the results.

Comment: The similar trends are supported by the analysis comparing calculated $\triangle$RBT with measured $\triangle$RBT.

The comment on the effect of alpine lakes is not well connected to the rest of the paragraph and should be expanded with some more detailed discussion.

Comment: We just hypothesize why Basel has such an alternating RBT. However, the RBT does not show a long term trend over the whole dataset. Finding the reason is not in the scope of this paper.

[Figure]

Eq 10 is dimensionally not consistent.

Comment: Thank you, we missed the density. Changed.

How did the authors select the periods in Table 6?

Comment: The start of the period is the time of the maximum heat input by NPPs at the respective station. We added this information to the text.

The authors could do the same calculation in continuous, for the entire period when the data are available (e.g., using the same two-year time windows as before).

Comment: This would be a good idea. However, tha aim was to use a time windows with the largest signal to noise ratio. Therefore we picked the largest ∆HI to avoid influences by short term trends.

P11, line 16: what is the BASF company? This should be explained.

Comment: We added two sentences to explain the BASF.

Why RBT in Figures 4 and 5 are different? How sensitive are the results of the correlation analysis to the filtering of the data?

Comment: Figure 5 has filtered RBT, cf. comment on the reviewer 1 on page 2

How the filtering parameters have been chosen and why 480 days has been used to shift the GDP-change time series? This number seems quite arbitrary.

Comment: It was shifted to ensure a visual match between the two data-sets (GDP and RBT). The shift can be explained by lagging and leading economic factors. This is explained in the manuscript. Mathematically the 480 days shift does not yield the largest positive correlation.

Appendices could be moved to the main text. In particular, the sentences in Appendix B should be revised because they have some syntax errors and typos. Figures A1 andA2 are inverted and the caption is the same. The analysis of parametersa2anda3should

be deepened and moved to the main text.

Comment: We move the biggest part of the appendix into the main text, as advised.

Technical corrections

P1, line 13: "but an" –> "but is an". Is "means of production" an appropriate term in thiscontext?

Comment: Thank you for the hint. We think means of production is appropriate.

P2, line 3 and following lines: the use of "Ta –> Tw" is informal and should be modified.

Comment: Thank you for your comment but we would like to keep it that way.

P2, line 8: "hydro-logical" –> "hydrological"

Comment: We changed it.

P2, lines 8-9: a reference is needed here.

Comment: This part has been moved and we added a reference in this sentence.

P2, line 16: is "revise" the most appropriate term here?

Comment: You are right. We use "test" now.

P2, line 20: "almost ideal" –> "ideal", "interesting", "meaningful"

Comment: Thank you, we changed it.

P4, line 13: "followed, by" –> "followed by". Please, thoroughly revise the punctuation-throughout the article (use of commas, missing close-brackets, etc).

Comment: We completely revised this part. The sentence is now rewritten.

P5, line 17: "ptovided" –> "provided"

Comment: We changed it.

P6, line 1: I would say that authors present fourTccalculations, not two.

Comment: We revised this part completely.

P6, line 18: "heat input by NPPsto the Rhine" –> "heat input by NPP to the Rhine"

Comment: We changed it.

P8, line 5: "(0.0350◦Cy−1)" –> "(0.0489◦Cy−1)"

Comment: We completely revised this table.

P10, line 15: "over the a time period" –> "over a time period"

Comment: Thank you, we changed it.

P11, line 1: "shorter timer scale but do not seem,to our" –> "shorter time scale but doC6 not seem, to our"

Comment: Thank you, we changed it.

P11, line 14: "A a discontinuity" –> "A discontinuity"

Comment: Changed. P11, line 19: "by a by a" –> "by a"

Comment: Changed.

References Benyahya L., Caissie D., St.Hilaire A., Ouarda T.B.M.J., Bobe B. 2007. A review of statistical water temperature models. Canadian Water Resources Journal 32: 179–192 Cai H., Piccolroaz S., Huang J., et al.2018. Quantifying the impact of the Three Gorges Dam on the thermal dynamics of the Yangtze River. Environ ResLett.;13(2018):054016. Caissie D., El Jabi N., Satish M.G. 2001. Modelling of maximum daily water temperatures in a small stream using air temperature. Journal of Hydrology 251: 14–28. Caissie D., Satish M.G., El-Jabi N. 2005. Predicting river water temperatures using the equilibrium temperature concept with application on Miramichi River catchments (NewBrunswick, Canada) Hydrol. Process. 19 2137–59 Gaudard, A, Weber, C, Alexander, TJ, Hunziker, S, Schmid, M. 2018. Impacts of using lakes

and rivers for extraction and disposal of heat. WIREs Water. 5:e1295. Gallice A., Schaefli B., Lehning M., Parlange M.B., and Huwald H. 2015. Stream temperature prediction in ungauged basins: review of recent approaches and description of a new physics-derived statistical model. Hydrol Earth Syst Sci 19:3727-3753 Isaak D.J., Luce C.H., Rieman B.E., Nagel D.E., Peterson E.E., Horan D.L., Parkes S.,Chandler, G.L. 2010. Effects of climate change and wildfire on stream temperature sand salmonid thermal habitat in a mountain river network. Ecol. Appl. 20, 1350–1371. Isaak D.J., Wollrab S., Horan, D., Chandler, G. 2011. Climate change effects on stream and river temperatures across the northwest U.S. from 1980-2009 and implications forsalmonid fishes. Climatic Change. 113: 499-524 Sahoo G.B., Schladow S.G., and Reuter J.E. 2009. Forecasting stream water temperature using regression analysis, artificial neural network, and chaotic non-linear dynamicmodels. J. Hydrol. 378 325–42 Sohrabi M.M., Benjankar R., Tonina D., Wenger S.J., and Isaak D.J. 2017. Estimation of daily stream water temperatures with a Bayesian regression approach. Hydrol.Process. 31, 1719–1733 Toffolon M., and Piccolroaz S. 2015. A hybrid model for river water temperature as a function of air temperature and discharge. Environmental Research Letters 10:114011 van Vliet M.T.H., Ludwig F., Zwolsman J.J.G., Weedon G.P., Kabat P. 2011. Global river temperatures and sensitivity to atmospheric warming and changes in river flow. WaterResour. Res. 2011, 47 Raman Vinna L., Wüest A., Zappa M., Fink G., Bouffard, D. 2018. Tributaries affect thethermal response of lakes to climate change, Hydrol. Earth Syst. Sci., 22, 31–51

---

## Author Response (AR2)

Blue: Authors comments

Report 1:

The manuscript has been improved substantially during the review process, as the authors considered most of the specific comments of both reviewers.
Thanks for the previous comments and the kind support.

A general comment by both reviewers was to expand the discussion of the results. However, a thorough discussion also with respect to the wider literature is still missing. Consequently, the manuscript still reads more like a case study report than a scientific article.
According to the reviewer's comments, we extended the aspect and hope to receive now an overall positive evaluation. We also added more references to previous work to put our findings in a broader context. In this manuscript we do propose a new method of regressing Tw vs Ta and, true, test this method in a case study.

A reorganization of the text so that methods and results are strictly separated (as suggested also in the previous round of reviews) is advised.
We moved the last paragraph of the introduction to the methods section. We also intensified the explanations in the method section. So, in our opinion there is a clear separation between introduction/methods/results.

Figure and table captions need to be revised so that they are distinct, concise, and self-explaining.
A thorough check of the manuscript syntax, grammar, spelling and punctuation is furthermore advised. I have included only a selection of the language issues I found in the specific comments.
As visible in the watemp_5_diff.pdf, caption- and formal revision has been undertaken.

Specific comments (page & line number according to track changes version):
Comments are also based on the watemp_4a_diff.pdf file.

**P2 L25 I would assume neural network based river temperature models are statistical models.**
Correct. We changed the wording accordingly and present neural networks as a subgroup of statistical models. "Artificial Neural Networks (ANN) are a subset of the statistical models and…"

**P2 L37 It is not clear that this sentence refers to the Markovic paper.**
Thanks. We changed it to: Using linear models, Markovic et al. (2013) show that between 81 % - 90 % of the Tw variability can be described by Ta. Furthermore, they show that 9 % - 19 % can be attributed to hydrological factors (e.g. discharge). The study was conducted for the Danube and Elbe basins using data from the 1939 to 2008. These two rivers have comparable discharge and catchment area to the Rhine river, which could mean his results are transferable to the Rhine river. These, although simple, linear models are able to clearly separate the different influences on Tw.

**P2 L48 Subsection heading necessary? (There is no 1.2)**
We deleted the heading.

**P3 L89 Column numbers needed?**
We deleted the column numbers in the text. The description of the columns is given in the caption.

**Tab 1 Omit "Lists of ". Description of columns in table caption not needed**

We deleted it.

**Fig 1 Revise caption. E.g. "Heat input by upstream NPP…"**
We changed the caption to: "Heat input by upstream NPP from 1969 to 2018 at each monitoring station."

**Tab 2 caption "conversion" instead of "coversion"**
Thanks for pointing out the typo.

**Fig 2-4 Use distinct figure captions or merge into multi-panel figure**
We changed the design and put figure 3 into a multi-panel figure as it shows a different weighing. Thanks for the comment.

**Fig 2 Add "NPP" to legend and add a legend item with "x" for the monitoring stations**
We added NPPs and Xs for all monitoring stations

**Tab 3 Omit "This table.."**
It is deleted.

**P11 L 217 Revise sentence**
The section was revised to:
"A catchment-wide hydrological flow model, estimating the flow speed at every grid point for every hydrological scenario, was not used. It had not been yet available for every grid point of the catchments and the focus of this study was to create a simple set-up, also transferable to other river catchments."

**P12 L 246 Revise sentence – How do grid points reach monitoring stations?**
We deleted the last part of the sentence as it is unnecessary. The temporal relationship is anyways explained in the next sentence using Δt:
"Tc (x0;y0; t0) was calculated by weighted (ACC _w) averaging Ta (x;y; t+_t (x;y)) over all grid points of the catchment area (x=1,…n y=1,…m) which was set by the measurement point (x0,y0). The time-lag dt was an estimate for the time it takes for a water droplet from a specific grid point (x,y) in the catchment area to the measurement location."

**Fig 6 Revise caption to make it more self-explanatory. Also: Relative contribution of what? What does "using by number" mean?**
We think the reviewer means Fig. 5 (in the new version it is Fig. 4). We changed the caption:
"ACC bins (x-axis) vs the relative contribution (y-axis). The grid points are binned by their ACC value. The red bars show the relative contribution (largest contribution normalized to one) by the number of grid points in this bin only. The white bars show the distribution using the number of grid points in this bin and weighing ACC _w."
The paragraph regarding this figure was also changed:
"The grid points were binned according to their ACC value. A high bin represents large rivers, a low bin their tributaries. The reason was to investigate the importance of the different ACC bins to the total Tc calculation. The ACC bin with the largest contribution in Fig. (4) was normalized to 210 one making it a relative contribution. The red bars (Fig. (4)) show the relative contribution (y-axis) of each ACC bin by the number of grid points in this bin only, no ACC _w weighing was applied. The results showed that the large number at low ACC bins (small water mass) have a larger influence compared to the rather low numbers at high ACC bins (e.g. large water masses, rivers, lakes). The difference in relative contribution is four powers of magnitude. The white bars show the relative contribution using the number of grid

points in the bin and the ACC_w weighing. 215 This distribution delivered rather equal importance to all grid points as it puts more weight on grid points covering lakes and rivers. The average difference in relative contribution is about 1 power of magnitude."

**Tab 4 Linear fits to what? No need to mention column numbers in caption. What does R2 > 1.99 mean (I assume it is a typo)**

We deleted the numbers in the caption. We also added that data-basis of the linear fit is described in the header. The "1.99" was a typo it should have been 0.19, Thank you, it was corrected.

**Tab 5 Would it be possible to call the first column "approach" or "model" rather than "descr."?**

You are right. "descr." is odd. We call it method, as we call it calculation method in the manuscript

**Fig 9 Omit "the three panels show". Omit "the" before Cologne**

Thanks. We changed that.

Once again many thanks for the support from reviewer 1, we appreciate the time spend on the manuscript and think that it has been improved by the comments provided.

Report 2 states

I believe that the authors improved the original version of the manuscript by clarifying some important aspects of their analysis.
Thank you for the time invested in proof reading, we to be able to further clarify and improve the manuscript with your support.

Still however I believe that there are some important issues that the authors should improve and clarify, including some requests of my previous review that were not addressed.
We added the full RMSE and NSC data for all calculated flow speeds to the supplement. The change in RBT is now continuously calculated (Fig. 6 in last Version). These are the main but not sole improvements regarding the robustness of our approach.

In general, grammar, syntax, and equation notation should be carefully checked throughout the manuscript and I suggest that the structure of some sections should be revised.
The manuscript was once again carefully proof read and also given to a third person just for proofreading.

I believe that the analysis and the quality of the manuscript can be substantially improved in several aspects, as suggested in my specific comments below.
Thank you once again for the constructive support; we addressed all points raised by you, below.

Introduction:

**- Line 20: I would not mention riparian vegetation together with meteorological forcing. Rather, I would move it to point 3, after opportune adjustments. In addition, I believe that, given the focus of the study, anthropogenic effects should be explicitly mentioned in a specific point.**
Even though we think that "riparian vegetation" does not fit perfectly there, we follow the reviewer advice and moved it to point [3] and mention the anthropogenic impact in the sentence before.

**- Lines 26-28: please revise this sentence: it seems that fluxes and boundary conditions are two distinct entities, while in many cases boundary conditions are fluxes. This sentence confirms that riparian vegetation should be removed from point 1 (it is not a flux).**
You are correct. We changed the sentence. The fluxes are now combined and the boundary and starting conditions as well, we wrote:
"A physical Tw model (Sinokrot and Stefan, 1993) usually parameterizes or estimates the meteorological and ground heat fluxes and adds anthropogenic heat input. Each modeled heat flux is then applied to the water mass, initialized with the starting and boundary conditions of source temperature and discharge. However, it is difficult to get a good estimation of these different 30 terms over a larger catchment area."

**- Line 29: does the term "parameter" refer to the fluxes mentioned in the previous sentence? If so, since fluxes are not parameters, I suggest using e.g., the word "terms".**
Thank you for the hint, we changed it accordingly to "terms".

**- Line 34: I do not understand the use of the term "analytic" here.**
We revised the sentence and hope it is clearer now:

"Using linear models, Markovic et al. (2013) show that between 81 % - 90 % of the Tw variability can be described by Ta. Furthermore, they show that 9 % - 19 % can be attributed to hydrological factors (e.g. discharge). The study was conducted for the Danube and Elbe basins using data from the 1939 to 2008. These two rivers have comparable discharge and catchment area to the Rhine river, which could mean his results are transferable to the Rhine river. These, although simple, linear models are able to clearly separate the different influences on Tw."

**- Line 37: please start a new paragraph after "... to the Rhine river."**
We did it and we also restructured the last part of the introduction.

**- Section 1.1: it is uncommon to have a subsection in the Introduction and I do not think that it is needed here, thus I strongly recommend to removing subsection 1.1. In doing so, a connection sentence between the first half of the Introduction and the second half will be required. As commented in my previous review, this second part of the Introduction reads more as a paragraph of Material and methods. I strongly recommend to fully revise the structure of this section focusing on presenting the objectives of the manuscript and an outline of the approach followed by the authors. Both points are not sufficiently addressed in the present version of the Introduction. Some specific comments are provided below:**
We moved the last part of the introduction to the methods part. Anything regarding the mathematical description of the model is in the methods part. We then rewrote the remaining last part of the introduction to connect to the methods. pls. cf. the watemp_5_diff file.

**\* Lines 57-59: this sentence is unclear and the link between the concept of Tc and the assessment of the impact of industry, meteorology and hydrology is confused and unsettled. The syntax should be revised (e.g., "combine ideas from")**
We restructured this part of the introduction and moved it to the methods part: The paragraph reads: "We investigated the change in anthropogenic heat input and its spatial and temporal heterogeneity along the Rhine combining 70 ideas from the spatial correlation models to develop a new method of calculating a representative catchment air temperature (Tc). Tc and discharge at the measurement station Q were used in a multiple linear regression Tc !Tw (Eq. 1). The resulting regression coefficients a1, a2 and a3 describe the magnitude of the respective influences (anthropogenic heat input, meteorological and hydrological)."

**\* Lines 61-62: I would not call the period 1979 to 2018 a "scenario"**
We changed it to "case".

**\* Lines 66-67: I do not agree with this sentence: Ta does not take into account the origin of water. This is particularly evident for ground water sources. This comment should be removed from the manuscript.**
We deleted the sentence. We think it was unclear too.

**\* Lines 70-72: this sentence is debatable since the hybrid model cited by the authors has been already used to evaluate the separate effects of anthropogenic and climate changes (Cai et al., 2018) and I have some concerns on affirming that a simple linear regression model (eq 1) can "allow for a clear distinction between meteorological, hydrological, and anthropogenic input".**
Maybe Cai et al., 2018 is not the best example, as they train their model Pre-Three Gorges Dam (TGD) and then apply it to the air temperature post-TGD. After that, they take the difference in simulated Tw and observed Tw. This approach provides estimates on the difference of a1 a2 and a3. These differences can be of anthropogenic origin.

Our model, if it were used on the TGD data-set, would only use the measured Tw data and interpreting the model parameters (a1), which delivers in our case the anthropogenic heat input. We would not investigate any possible anthropogenic or natural changes to the meteorological influence or the influence of discharge. We also do not use a subset of our data for training and then relate it to others but stepwise compare sub-datasets in a time series.

However, we agree with the reviewer, using a1 from the air2 stream model (Marco Toffolon and Sebastiano Piccolroaz 2015 *Environ. Res. Lett.* 10 114011) would also show the anthropogenic change. The simple air2stream mode with 3 paramteres from Toffolon et al 2015 is basically the same model that we use.

Hence, we deleted the sentence and added a thought to the introduction:

"Hybrid models can reproduce river water temperatures better than simple statistical models (linear regression) \citep{Tof2015}. The approach includes more parameters and thus, is more complex. A simple hybrid model with for example, three parameters, is comparable to a statistical model with the same number of parameters."

**\* Figure 2 (the study site) should be probably cited here.**
We added an additional legend.

**- As pointed out in the first revision round, I believe that the Introduction would benefit from mentioning existing literature on the assessment of anthropogenic impact on river water temperature (e.g., Cai et al., 2018; Gaudard et al., 2018; Råman Vinnå et al., 2018, just to cite some recent papers).**
Included.

**Section 2.1.**
**- Line 83-84: "reference" --> "data provider"/"data source" ?**
Was changed to data-provider.

**- Lines 90-92: please, check the verb tenses. The use of "by us" is not recommended.**
We deleted "by us" and changed the tense. Generally, we use active voice as it is now recommended by many chief editors:
"Nature journals prefer authors to write in the active voice ("we performed the experiment...") as experience has shown that readers find concepts and results to be conveyed more clearly if written directly. We have also found that use of several adjectives to qualify one noun in highly technical language can be confusing to readers. We encourage authors to "unpackage" concepts and to present their findings and conclusions in simply constructed sentences." Nature Editor in Chief
Of course using passive voice and then adding by us is not helping to generate a fluently readable text and therefore we deleted it.

Section 2.3.
**- As commented in the first revision round, the sentences in the first part of this section are too fragmentated and short.**
**The authors should explain in detail how they aggregated the heat input due to each NPP to obtain the overall heat input at each gauging station.**
We revised the beginning and increased the readability. We added an equation explaining the heat input calculation

Section 2.4

**- This section is disconnected from the previous, since the authors did not mention the use of GDP in the preceding part of the manuscript. This is something that should be mentioned in the Introduction, where the authors should properly (i.e., concisely but clearly and exhaustively) introduce objectives and approaches of their study.**

In the last paragraph of the introduction we added now extra lines to explain our objectives and the use of GDP data and the data from nuclear power plants.

"Short term economic changes, observable in the change of GDP, may influence Tw on shorter time scales (<5 years). As several industrialized hotspots are present along the river, this impact might be heterogeneous. Using the nuclear power production and GDP data, we also investigated the heterogeneous anthropogenic impact on Tw along the Rhine at four monitoring stations (Basel, Worms, Koblenz, and Cologne)."

**- Figure 2: I would present the catchment closed at the most downstream gauging station. In this way the entire region analyzed in the study would be presented. Please substitute the symbol used for the Fessenheim NPP as it can be confused with a gauging station.**

We changed the symbol for Fessenheim. Presenting the catchment area of Cologne would indeed show the entire region but one would also loose details. So we decided that it is best decision to show Koblenz.

**- Line 140: The use of subsections, subsubsections (e.g., 2.3.1) and paragraphs (e.g. Accumulation) is not harmonized. Please check it throughout the manuscript.**

Every subsubsection is numbered, if there is more than one in a subsection. If not is just a separate heading without numbering was included.

**- Line 142: please revise the syntax ("... the the ..."; "... this very grid point")**

We changed the total paragraph. The part is now in section 2.6.

"Additionally, the accumulation number ACC was obtained from the data-set. It defines how many cells in total were draining into a particular cell and it is a measure for the size of a river. Finally, a grid, which defines the catchment area, the ACC and the hydrological distance was established spanning the whole catchment area. Figure (2) shows the catchment area, the hydrological distance and the calculated flow time to the Koblenz monitoring station. The ACC displays is the number of grid points which were hydrologically connected to this specific grid point. Figure 3 (top panel) shows the distribution of the ACC. Large rivers, which have a large ACC number, such as the Rhine, Main, Neckar are easily visible due to their green to yellow color."

Section 2.7

**- In my view, part of subsection 1.1 (including equation 1) should be moved here. In the Introduction the authors can easily comment on their approach without showing the equation. Indeed, as commented above, in the Introduction the authors should focus on objectives and approaches used in their study in a concise and clear way.**

We followed the reviewer's advice and moved the subsection 1.1. to the methods part.

**More important, as commented in my previous revision, I do not agree that "The offset a1 (RBT) combines all other influences, which are controlled by anthropogenic sources". a1 accounts to much more than solely anthropogenic sources as it summarizes all contributions that are not directly linked**

to Ta and Q (groundwater inflow, geothermal flux, vegetation shading, tributary heat flux, upstream heat flux, etc). Please see Segura et al (2015) for an useful overview on the physical meaning of the intercept a1. Still, the authors can suppose that if all the aforementioned conditions are kept unchanged, changes in a1 can be linked to changes in anthropogenic sources (as they comment at lines 267-270). This is conceptually and formally different. The authors should carefully avoid any misunderstanding around the meaning of a1 (adjust also Introduction, lines 265-, Conclusions and all sentences where the meaning of a1 is commented).

The reviewer comment is absolutely correct. The sentence "The offset a1 (RBT) combines all other influences, which are controlled by anthropogenic sources" is not precise and leads to misunderstanding. It was corrected, that anthropogenic changes which effect meteorology (change in shading for example) are represented in a2. Any anthropogenic changes going together with discharge would change a3. This is now clarified. We changed the wording.

We already state in the introduction of the Methods:
"Using the multiple regression (Eq. 1), we especially investigated the change of a1 over time, which we call in this study the Rhine base temperature (RBT). This temperature represents Tw without the influence of meteorology (Ta) and discharge (Q). RBT was defined to be an indicator for industrial heat input and the use of Rhine water as cooling agent, in case both are mostly independent of Ta and Q."

The overall industrial and power production would have to be dependent on Ta and Q to have an effect on a2 and a3. During extreme Q and Ta this may happen, but is very unlikely and very seldom the case. To avoid any problems with this, we always regress a two years period, because it is unlikely that over a period of two years production strongly correlates with air temperature or river discharge. This is now mentioned in the manuscript.

The lines 267-270 are now deleted.

We went over all parts where a1 is mentioned and revised the phrasing to be more consistent and to follow the suggestions. a1 is now always labeled as anthropogenic heat input, which is more precise than being the total anthropogenic influence.

- Line 50: "over the whole" --> "over the whole catchment"?
Thanks, we changed it

- Equation 4: I would have expected to see "Tc(t0-Delta t(x,y))" and "Q(x0,y0,t0)". Is this correct? As for the first point: since, as far as I understand, Tc is spatially averaged it should not depend on x and y. In addition, to predict Tw at time t0 at the closing section, Tc should be taken at time t0-Delta t(x,y) for the i-th cell located in (x,y) to account for the time delay due to water routing. Considering t0+Delta t is coherent with eq 8, where the authors define Delta t as a negative number, but this comes later in the text and in my view is misunderstandable.

We changed the equation in such a way that Tc(x0, y0, t0+Δt(x,y). Tc is now connected to x0,y0 which is the measuring point and sets the extension of the catchment area.
We also added a small explanation on the time lag being negative and hope we made it better understandable now. We also made the appropriate changes to all other equation if necessary.
Good point. Thanks.

- Equations 5 and 6: similarly to above, the time lag is not correctly accounted for: "T(x0,y0,t0) = Ta(x0,y0,t0 -Delta t)" and in the second equation the integral should be from i=t0-Delta t to i=t0. Delta t can be confused with the time lag due to routing while here it refers to the lag due to water inertia. Moreover, here and in the following equations "T" is not defined. The authors should avoid any misunderstanding, be precise in the use of notation, and carefully check all equations.
We changed the equations accordingly.

Inertia time lag vs advection time lag:
You are right there are two time lags. One is a constant (inertia) the other one depends on distance and flow speed. We state in Line 187 (newest version)
"The new _t (x;y) represents the mismatch by advection but not specifically the mismatch through thermal inertia. The thermal inertia would be independent of s(x;y) and a constant added to _t. However, we are of the opinion, that a sufficient part of the thermal inertia time-lag was included in our representation of _t (x;y)."
We also added a sentence about the inertia in relation to flow speed at
Line 203 (newest Version)
"The reason for the small flow speed (0.4 ms-1) and the even smaller ones at Basel and Worms could be the thermal inertia. As thermal inertia is not explicitly 205 represented in dt (Eq. 9) a smaller flow speed could compensate for that, especially in smaller catchment areas."

**- Line 165: "reason reason". There are several other points where the syntax should be checked. I will not comment further on this, but strongly encourage the authors to thoroughly check the entire manuscript.**
Thank you and we are sorry about the errors. We proof read manuscript again.

**- Lines 165-176: this part is confused. This is what has been already said in eq 4 and the reference to the 8-day lag is unclear and probably unnecessary: from the text one understands that only the routing lag has been considered, while the time lag due to thermal inertia has not. Overall, this whole section requires significant improvement and clarification.**
We agree to a certain extend. The thermal inertia is represented in our time lag, to a certain extend. We changed the paragraph starting Line 175 (newest Version). There we distinguish between thermal inertia and advection.
We also added a sentence about the inertia in relation to flow speed at
Line 203 (newest Version)
"The reason for the small flow speed (0.4 ms-1) and the even smaller ones at Basel and Worms could be the thermal inertia. As thermal inertia is not explicitly 205 represented in dt (Eq. 9) a smaller flow speed could compensate for that, especially in smaller catchment areas."

**- Lines 183-186: the authors should better explain how the flow speed has been estimated. Do they mean RMSE between observed and simulated Tw using the Time lag + weight model (or just considering the Time lag, or Time lag + weight + ACC)? Is this value of flow speed confirmed also when analyzing the other gauging stations? The authors should show (at least in the supplementary material) the relationship between RMSE and flow speed for all gauging stations. Is the minimum in RMSE clear and unequivocal? Did the authors optimize independently the flow speed when testing the different definitions of Tc?**
We followed the advice, added the analysis of all stations to the supporting material and hope to make it better understandable. We also added that the calculation of RMSE there was done using the whole data-set.

**- Table 4: "which are statistical significant only if R2 > 1.99" there should be an error here. Please, specify that R2 refer to the linear trends. As suggested in my previous revision, I believe that the authors could add the Pearson coefficient between Ta and Tw to further (and more robustly) support their reasoning.**

This is a typo. comment by reviewer1. Thanks. The correct number is 0.19. We added the Pearson coefficient to the table.

**- Lines 238-241: this paragraph is chaotic. Some concepts are repeated and not necessary in this context (e.g., in Base Ta and Tw show similar behavior).**
We changed the whole paragraph.

**Section 3.2**
**- Line 246: "catchment-wide Delta t" the use of catchment-wide is probably not appropriate here as it could be understood as a constant Delta t for the entire catchment.**
We deleted it.

**- Line 250: see my comment relative to Lines 183-186. Here and below: NCS-->NSC**
We added the calculation method of RSME and NSC to the caption and it is as well in the paragraph.

**- Line 255: I would not say that a figure is the reason of the results, but that the content of the figure can explain the results.**
Correct sorry, for that. We changed it.

Section 3.3
**- Lines 275-276: actually RBT seems to decline some years in advance (in 1995 looking at the running mean in Fig. 7). Can the authors add a comment on this?**
Correct. Beginning 1993 there was a recession and a crisis in the trade balance, which might have affected the long term trend. We think that this was a coincidental trigger for this decline.
We wrote:
"The RBT started its decline a 1-2 years before 1995, which might have been triggered by the recession in 1993 and a sharp drop in the German trade-balance."

**- Table 6: here the authors do not show that the two series of Delta RBT have "similar trends" since the results in the table only depend on initial and final values (which has clear side effects and limitations). To test if the trends are similar, they should calculate Delta RBT in continuous with eq 2 and using the time series of the heat input, and compare it with the time series of RBT shown in Figure 7. This is something that I already suggested in my first review and that I strongly recommend adding to the analysis. While the authors replied that they wanted to pick the largest Delta HI to avoid influences by short term trends, I believe that analyzing the time series is needed to properly show that the two time series have comparable medium- to long-time trends (short term trends can be easily filtered e.g., with a moving average if required).**
The time series of ΔRBT was added to Figure 4. We added also an analysis to the paragraph, stating that the time series of ΔRBT at Fig. 4 follows the RBT time series and the HI by NPPS. It is interesting that before 1995 an offset between RBT and ΔRBT occurs. We think that between 1985-1995 the NPP Power production stayed constant but the GDP increased by 30% during this period. So the input from industrial production seems to have contributed to this offset.

**- Lines 285-317: as commented above, the term RBT summarizes several contributions besides anthropogenic effects. This should be properly recalled in this section, since this is the most likely reason of the differences between RBT and GBT trends (including the specific cases commented by the authors), although I agree that a significant correlation is visible in figure 9.**

We added a sentence to the top paragraph of section "Short term trend", reminding the readers, that RBT is not only influenced by industrial production (GDP) but also other sources, which we are not investigated in this paper.

**As commented in my previous review, the comment on the effect of lakes is somehow disconnected from the rest of the paragraph. I perfectly understand and agree that "finding the reason [of the peculiar trend in Basel] is not in the scope of this paper", but the sentences at lines 289-293 should be better contextualized. The reasoning on stratification is fine, the fact that deep water temperature is somehow decoupled from Ta is fine as well (but only for deep lakes), however this does not explains the trend shown in Basel nor the reader knows if such lakes contribute to the Rhine through surface water (natural lakes) or deep water (hydropower reservoirs with deep intake). Better than trying to draft a possible explanation without effectively providing the proper information and hypothesis, would be to simply write that the trend in Basel cannot be explained in this analysis.**
Thank you for the comment, we deleted the hypotheses.

Conclusions:
**- Line 340: the term "reanalysis" and "forecast" are probably not the most appropriate here.**
We remove reanalysis and forecast

**- Line 342: fluxes are not parameters. I would say that fully physical models requires all meteorological data in input.**
We changed it from parameters to input.

**- Lines 343-348: the comment on tropical and subtropical rivers is somehow disconnected from the study presented by the authors and personally I do not believe that it is appropriate here. I do not believe that the study by Morril et al (2005) suggests that "this case study of the Rhine can be applied globally". On the contrary the comment on the possible coupling with catchment-wide hydrological models is more significant but poorly examined (please improve it, adding a comment also on the limitations of the analysis).**
We deleted the sentence about tropical rivers. We do think that Morril et al 2005 gives a hint to applicability of out model. Generally for our model to work, just a linear relationship between Ta and Tw is need. The information provided by Moril et al 2005 is that they can reasonably establish a linear relationship and their RMSE is in the same range as ours and therefore we are confident, that it is possible to apply this to other rivers. We did a quick survey (not in this manuscript) of the Elbe and can reproduce the industrial development there using a1, too.

References included.

[revised manuscript text omitted]

---

## Author Response (AR3)

Dear editor,

Dear publishers,

Thank you very much for your help and support. We would like to thank the editor for taking over the review process at such short notice.

**Line 104 „a the" – please check**

Thank you, we removed "a"

**Line 147 personal communication – with who?**

We added: „with technicians from NPPs"

**Lines 34, 473, 497 I would prefer "physical based" to "physical" model**

We changed it.

**Line 324 "a lot of warming" – rephrase**

We changed it to "most warming"

**Line 350 should be "in bold"**

We changed it

**Line 495 "may be explored" rather than "will need to be proven"**

Thank you, we changed it

**Line 507 "more integrative" in what respect? Including river temperature management?**

We removed the sentence.